# MixSignGraph: A Sign Sequence is Worth Mixed Graphs of Nodes

**Shiwei Gan**[†]    **Yafeng Yin**[†*]    **Zhiwei Jiang**[†]    **Lei Xie**[†]    **Sanglu Lu**[†]    **Hongkai Wen**[‡]

[†] State Key Laboratory for Novel Software Technology, Nanjing University, China
[‡] Department of Computer Science, The University of Warwick, UK
{sw,yafeng,jzw,lxie,sanglu}@nju.edu.cn   hongkai.wen@warwick.ac.uk

## Abstract

Recent advances in sign language research have benefited from CNN-based backbones, which are primarily transferred from traditional computer vision tasks (*e.g.*, object detection, image recognition). However, these CNN-based backbones usually excel at extracting features like contours and texture, but may struggle with capturing sign-related features. To capture such sign-related features, SignGraph model extracts the cross-region sign features by building the Local Sign Graph (LSG) module and the Temporal Sign Graph (TSG) module. However, we emphasize that although capturing cross-region dependencies can improve sign language performance, it may degrade the representation quality of local regions. To mitigate this, we introduce MixSignGraph, which represents sign sequences as a group of mixed graphs for feature extraction. Specifically, besides the LSG module and TSG module that model the intra-frame and inter-frame cross-regions features, we design a simple yet effective Hierarchical Sign Graph (HSG) module, which enhances local region representations following the extraction of cross-region features, by aggregating the same-region features from different-granularity feature maps of a frame, *i.e.*, to boost discriminative local features. In addition, to further improve the performance of gloss-free sign language task, we propose a simple yet counter-intuitive Text-based CTC Pre-training (TCTC) method, which generates pseudo gloss labels from text sequences for model pre-training. Extensive experiments conducted on the current five sign language datasets demonstrate that MixSignGraph surpasses the most current models on multiple sign language tasks across several datasets, without relying on any additional cues. Code and models are available at: https://github.com/gswycf/SignLanguage.

## 1   Introduction

Advancements in computer vision (CV) and natural language processing (NLP) technologies have significantly facilitated the development of sign language (SL) research, including Sign Language Recognition (SLR) and Sign Language Translation (SLT). The SLR task encompasses Isolated SLR (ISLR) (1) and Continuous SLR (CSLR) (2; 3; 4), aiming to recognize isolated or continuous signs as corresponding glosses or gloss sequences. While the SLT task focuses on translating continuous signs into spoken language (5). In the current SL framework, the standard approach first uses 2D/3D CNN-based backbones to extract visual features (6; 7; 8), then employs temporal modules (5; 9; 10) to capture dynamic changes in sign frames, and finally adopts a Connectionist Temporal Classification (CTC) decoder to obtain the gloss sequence or a translation model to generate the spoken sentence.

Unlike other CV tasks where contour and texture representations are crucial, SL tasks need to focus on both manual and non-manual features, particularly the collaboration of these cues in different

---

[*]Yafeng Yin is the corresponding author.

39th Conference on Neural Information Processing Systems (NeurIPS 2025).

regions (11; 12), while the traditional CNNs may fail to capture the collaboration of signs across different regions, thereby limiting their ability to extract effective sign features. To enhance the CNN-based backbone's ability to extract sign-related features, existing work has focused on designing various CNN-based backbones by incorporating domain knowledge, such as skeleton information (9), depth images (13), or local areas (10). Additionally, some studies have adopted different training strategies *e.g.*, back translation (14), knowledge distillation (15) and contrastive learning (6). Besides, in SLT tasks, many current SOTA models also rely on gloss annotations to pre-train their backbones with CTC loss, aiming to improve the SLT performance. When gloss annotations are missing, *i.e.*, in Gloss-Free SLT (GFSLT) tasks, how to improve the SLT performance is still a challenging task.

To address the limitation of CNN-based backbones, SignGraph (4) transforms SL frames into graph nodes and proposes the Local Sign Graph (LSG) and Temporal Sign Graph (TSG) modules to model spatial and temporal cross-region dependencies. However, such strategies may inadvertently degrade fine-grained local features due to the emphasis on cross-region aggregation. Regarding improving the gloss-free SLT performance, the existing models have attempted to introduce attention mechanisms (16), contrastive language-image pre-training (17) or vector quantization (18).

Building upon prior work, we propose a simple yet effective sign graph neural network, MixSignGraph, for SL tasks (CSLR and SLT). In addition to extracting cross-region features, MixSignGraph introduces a Hierarchical Sign Graph (HSG) module to enhance local representations from feature maps of different granularities, mitigating the degradation of local features that may occur during cross-region modeling in SignGraph. The main intuitions are: Cross-region feature extraction may lead to the local representation of the current node being diluted by features from other regions, resulting in the degradation of local information. Moreover, the content within a single region can be represented at different levels of granularity through hierarchical feature maps generated by down-sampling. Therefore, we propose a Hierarchical Sign Graph (HSG) module, which connects corresponding regions across multi-granularity feature maps. This hierarchical design allows the model to reinforce local representations by aggregating complementary information from different granularities, thereby mitigating the degradation caused by cross-region feature fusion. Besides, to further improve the performance of sign language tasks without gloss annotations, we propose a simple yet counter-intuitive Text-based CTC Pre-training (TCTC) method, which generates pseudo gloss labels from text labels for model pre-training. We make the following contributions:

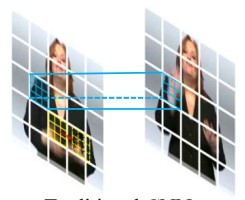
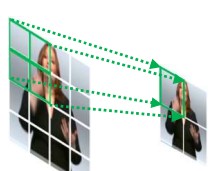

Traditional CNN: in spatial, temporal dimension

Traditional downsampling: from high to low resolution

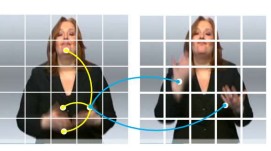
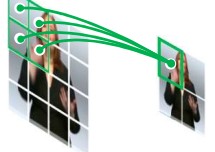

LSG/TSG: across regions in one or adjacent frames

Our HSG: the same regions across feature maps

Figure 1: (Left) Comparison between CNNs and LSG/TSG in SignGraph. (Right) Message passing of the same regions among different-resolution feature maps between downsampling in CNNs and our HSG.

- To address the limitations of SignGraph (4), we propose a simple yet effective sign graph neural network, called MixSignGraph. In addition to extracting cross-region features, MixSignGraph introduces a Hierarchical Sign Graph (HSG) module, which enables bidirectional exchanges of feature maps with different granularities, to enhance local representations from hierarchical feature maps. The proposed HSG can mitigate the degradation of local features that may occur during cross-region modeling.

- A simple, effective, yet counter-intuitive training method named Text-based CTC Pre-training (TCTC) is proposed for the GFSLT task, which generates pseudo gloss annotations based on text labels. The proposed TCTC effectively improves the performance of GFSLT and narrows the gap with gloss-based SLT models.

- The extensive experiments on common SL tasks (including CSLR and SLT) over five public datasets demonstrate the superiority of the proposed MixSignGraph, which shows promising results and does not use any extra cues.

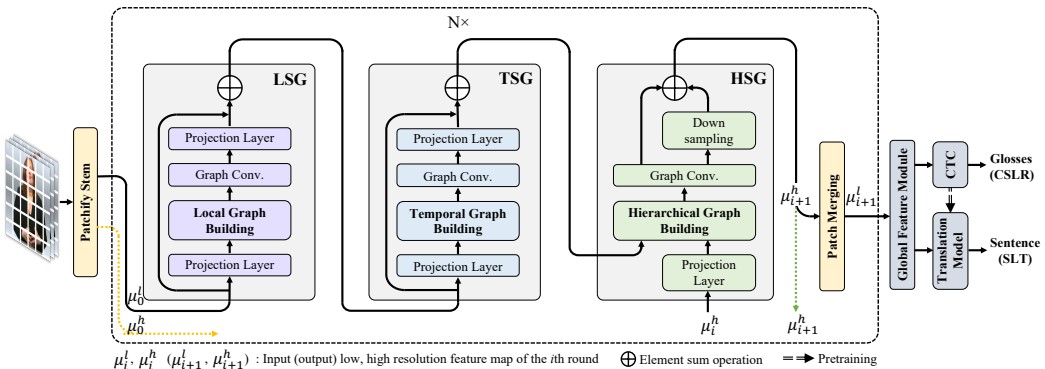

Figure 2: The proposed MixSignGraph architecture.

## 2 Preliminaries

For a SL sequence $f=\{f_i\}_{i=1}^{\theta}$ with $\theta$ frames, the target of CSLR task is to get a recognized gloss sequence $g = \{g_i\}_{i=1}^{\vartheta}$ with $\vartheta$ glosses, while the target of SLT task is to generate a spoken language sentence $t = \{t_i\}_{i=1}^{\varsigma}$ with $\varsigma$ words based on input $f$. SignGraph (4) introduces the following modules to model the SL sequence $f$:

**Patchify Stem.** To apply graph convolutional networks (GCNs) directly to SL frames instead of skeleton data, a patchify stem is employed to divide each RGB frame with height $H$, width $W$ into $N = HW/P^2$ patches (or nodes) $v_i = \{v_{ij}\}_{j=1}^{N}$, with the corresponding feature embedding $\mu_{ij} \in \mathbb{R}^D$. Here, $v_{ij}$ represents the $j$-th patch in the $i$-th frame $f_i$, $D$ denotes the feature dimension, and $P$ is the patch size.

**Local Sign Graph Learning.** To capture correlations between different spatial regions, such as the face and hands, SignGraph (4) introduces a Local Sign Graph (LSG) module to extract intra-frame cross-region features. Specifically, it computes the feature distances between patches within each frame using a distance function $\mathcal{DIS}$ (19). Based on these distances, a K-nearest neighbor (KNN)-based graph construction strategy is used to obtain local sign graph, and graph convolution is applied to integrate features across nodes, enabling effective modeling of intra-frame spatial dependencies.

**Temporal Sign Graph Learning.** To capture the movements of the body, hands, and face, SignGraph (4) introduces a Temporal Sign Graph (TSG) module that dynamically establishes connections between regions across consecutive frames, enabling the learning of inter-frame cross-region features. Similarly, it calculates feature distances between patches in adjacent frames and adopts a KNN-based graph construction strategy to effectively model cross-region features between frames.

**Patch Merging.** Considering that patches with a fixed window may not effectively capture sign features (*e.g.*, hand regions may be split by different patches), SignGraph (4) adopts the patch merging module to downsample feature maps by a factor of 2, to generate larger-size patches.

## 3 Method

**Overall Framework.** As shown in Figure 2, when given the video frames, a patchify stem is adopted to convert each frame into a set of patches (*i.e.*, nodes) $v_i = \{v_{ij}\}_{j=1}^{N}$. Then, the three key modules are utilized to construct graphs and capture sign-related features as follows. First, by adopting the LSG and TSG modules, MixSignGraph can learn both the correlation of cross-region features within one frame and the interaction of cross-region features among adjacent frames. Second, by aggregating the same-region features from different feature maps of a frame, we build a hierarchical sign graph $G_i^h$, to bidirectionally exchange features and mine different-granularity one-region features. After that, we adopt a global feature module (2; 6) to learn global changes of whole frames. To tackle CSLR tasks, a classifier and a widely-used CTC loss (20) are adopted to predict the probability $p(g|f)$ of the target gloss sequence. In SLT tasks, we adopt a translation model to convert feature sequences or gloss sequences into text sequences. To further improve GFSLT performances, we propose the Text-based CTC Pre-training (TCTC) mechanism, which generates pseudo gloss labels from text sequences for model pre-training.

**Hierarchical Sign Graph Learning.** Despite the benefits of capturing cross-region features for improving SL performance, it may degrade the representation quality of local regions. Besides, traditional CNNs can only downsample one region from coarse to fine representations, lacking the ability to facilitate interaction between different-granularity feature maps. To address this issue, we introduce the HSG module, which enhances local information from feature maps of different granularities after cross-region feature extraction. In this way, we can combine hierarchical features of the same region and make the

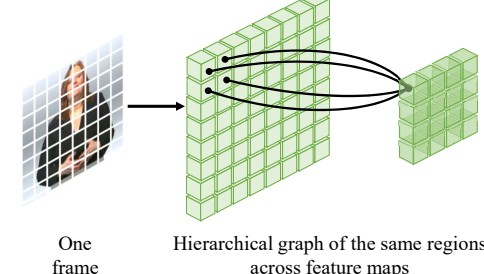

One frame — Hierarchical graph of the same regions across feature maps

Figure 3: Graph construction in hierarchical sign graph module.

model focus on both global shapes and detailed local features of a region at the same time. As shown in Figure 3, given two feature maps $\mu_i^h \in \mathbb{R}^{[H^h \times W^h \times C^h]}$ in high resolution and $\mu_i^l \in \mathbb{R}^{[H^l \times W^l \times C^l]}$ in low resolution for frame $f_i$, we have $N^h = H^h W^h$ nodes $v_i^h$ for $\mu_i^h$ and $N^l = H^l W^l$ nodes $v_i^l$ for $\mu_i^l$. Here, $H^h, W^h, C^h, H^l, W^l$ and $C^l$ are the height, width and the number of channels of $\mu^h$ and $\mu^l$ respectively, and $\frac{H^h}{H^l} = \frac{W^h}{W^l} = s > 0$, where $s$ is the downsampling factor. As shown in Figure 2 and 3, to construct the hierarchical sign graph, we first use a projection layer with weights $\Theta_1^h$ to map $\mu_i^h \in \mathbb{R}^{[H^h \times W^h \times C^h]}$ to $\mu_i^{h'} = \mu_i^h \Theta_1^h$, ensuring the same dimensionality as $\mu_i^l$, where $\mu_i^{h'} \in \mathbb{R}^{[H^h \times W^h \times C^l]}$. Then, we add an undirected edge $e(v_{ij}^h, v_{ik}^h)$, if the node $v_{ij}^h$ in $\mu_i^{h'}$ and the node $v_{ik}^l$ in $\mu_i^l$ correspond to the one region.

$$e_i^b = \{e(v_{ij}^h, v_{ik}^l) \mid j \in [0, N^h)\}, \quad k = \left( \left\lfloor \frac{\left\lfloor \frac{j}{W^h} \right\rfloor}{s} \right\rfloor \times W^l + \left\lfloor \frac{j \% W^h}{s} \right\rfloor \right) \tag{1}$$

In this way, we can get the edge set $e_i^b$ between $\mu_i^l$ and $\mu_i^h$ for the frame $f_i$, and obtain the hierarchical sign graph $G_i^H = \{(v_i^{h'}, v_i^l), e_i^b\}$. After that, we apply a graph convolutional layer $\mathcal{GCN}_H$ to enable bidirectional exchanges of the same regions across different-granularity feature maps to perform feature fusion.

$$\mu_i^h, \mu_i^l = \mathcal{GCN}_H(\{\mu_i^{h'}, \mu_i^l\}, e_b^i), \quad \mu_i^l = \mu_i^l + \mu_i^{h'} \Theta_2^b \tag{2}$$

Unlike LSG and TSG, which dynamically build edges based on the KNN algorithm to capture cross-region features, the HSG focuses on fusing features belonging to the same region at different granularities, aiming to enhance local features of one region.

**Mixed Sign Graph Learning.** Based on the previous work SignGraph, we insert the proposed HSG module after the LSG and TSG modules to enhance local information from feature maps of different granularities. For convenience, we represent the $i$th $LSG, TSG$ and $HSG$ module as $LSG_i, TSG_i$ and $HSG_i$, respectively.

## 3.1 Text-based CTC Pre-training

**Gloss-based Tasks.** For CSLR tasks, CTC loss with gloss annotations has become the de-facto loss function in current CSLR models. While in gloss-based SLT tasks, the current SOTA SLT models (22; 7) rely on pre-training, in which CTC loss with gloss sequences is used to optimize the backbone to extract effective video-level features, as described in Equation 3. After the pre-training, these SLT models further introduce a pre-trained translation model (*e.g.*, mBART (23)) and fine-tune the translation model for SLT, as defined in Equation 4. Here, $\mathcal{L}_{CTC}$ denotes CTC loss function, $\mathcal{L}_{CE}$ denotes cross-entropy loss function, $\hat{g}$ and $g$ respectively denote recognized gloss sequence and labeled gloss sequence, $\hat{t}$ and $t$ respectively denote translated sentence and labeled sentence, $\Theta_R$ and $\Theta_T$ denote the parameters of recognition and translation models.

$$\min_{\Theta_R} \mathcal{L}_{CTC}(\hat{g}, g) \tag{3}$$

$$\min_{\Theta_R, \Theta_T} (\mathcal{L}_{CTC}(\hat{g}, g) + \mathcal{L}_{CE}(\hat{t}, t)) \tag{4}$$

**Gloss-free Tasks.** The above pre-training process enables the backbone to learn segmentation and semantic information from gloss annotations, which are crucial for improving the SLT performance.

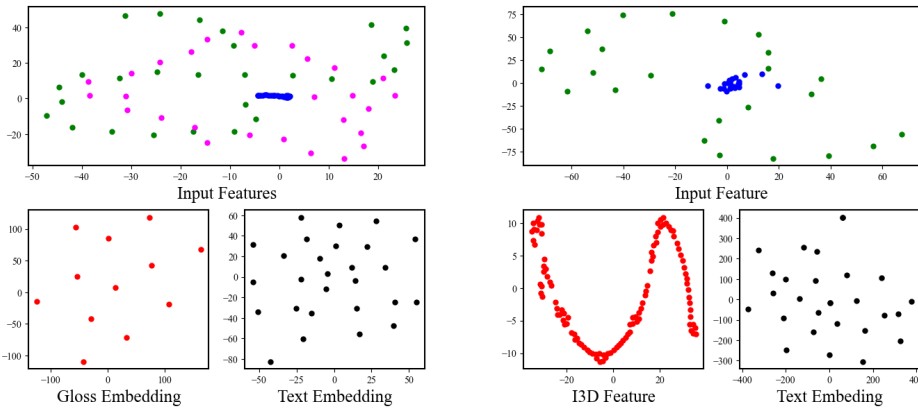

(a) Visualization of feature distribution of one test sample in PHOENIX14T dataset.

(b) Visualization of feature distribution of one test sample in the How2Sign dataset.

Figure 4: Visualization of feature distribution, which is displayed via t-SNE (21). Blue, green, and pink points represent the input features of the translation model by NOT using TCTC, using TCTC, using CTC pre-training with gloss annotations.

However, in the other SLT tasks, *i.e.*, gloss-free SLT tasks, there are no gloss annotations, since annotating glosses is a labor-intensive task.

**Text-based CTC Pre-training (TCTC)**: To improve the performance of these gloss-free SLT tasks, we propose a simple, effective, yet counter-intuitive training method called Text-based CTC Pre-training (TCTC), which generates pseudo gloss sequences from labeled sentences for pre-training. Specifically, in TCTC, we obtain the pseudo gloss sequence $t'$ by simply pre-processing labeled sentence $t$, including removing punctuations, lemmatization and word-level tokenization. Then, we directly use the CTC loss with pseudo gloss sequence $t'$ to optimize the backbone during initial pre-training, and then fine-tune the entire model for translation, as depicted in Equation 5, 6.

$$\min_{\Theta_R} \mathcal{L}_{CTC}(\hat{t}', t') \tag{5}$$

$$\min_{\Theta_R, \Theta_T} (\mathcal{L}_{CTC}(\hat{t}', t') + \mathcal{L}_{CE}(\hat{t}, t)) \tag{6}$$

**The principle of TCTC**: Typically, CTC loss requires that the source sequence and the target sequence are aligned in the same order. Thus, CTC loss is often used in the CSLR task, where the sign video and gloss sequence have the same order. However, in SLT tasks, the alignment between the SL video and the text sequence is usually non-monotonic, due to the difference in grammatical rules between SL and spoken language. To tackle the challenge of GFSLT task, TCTC generates pseudo gloss sequences from labeled sentences and then adopts CTC loss for model pre-training. Although the generated pseudo gloss sequences from text/sentences may be different from the ground-truth gloss sequences, our TCTC is still crucial for improving SLT performance. The two possible reasons can be concluded as follows. First, given the discrete nature of languages, the inputs and outputs of translation models (*e.g.*, mBART used in SLT) are in a categorical, discrete-valued space (*i.e.*, typically as token embeddings (24)). Second, without CTC pre-training, the features of adjacent frames/clips (extracted by the backbone) tend to remain in a continuous-valued space, while CTC pre-training with gloss sequences guides the model to segment video features into discrete units. Therefore, by using CTC with pseudo gloss sequence $t'$ to pre-train the backbone, TCTC provides a weak tokenization function, even if it may result in inaccurate segmentation.

To verify the effectiveness of TCTC, we visualize the feature distributions input to the translation model, *i.e.*, the features output from the CSLR model, while using or NOT using TCTC. As shown in the bottom left part of Figure 4, the embedding distributions of gloss sequence (*i.e.*, red points) and words in spoken sentence (*i.e.*, black points) are normally in a discrete-valued space, as described in the first reason of the previous paragraph. When moving to the top left part of Figure 4, the features extracted by NOT using TCTC are located at close positions (*e.g.*, blue points on a continuous curve), while the features extracted by using TCTC scatter at different positions (*i.e.*, green points in a discrete-valued space), as described in the second reason of the previous paragraph. It demonstrates

that introducing CTC loss with pseudo gloss annotations can guide the model to extract discrete and tokenized video features. From the perspective of feature distribution space, the pattern of feature distribution by using TCTC is similar to that of word embedding in spoken sentences, *i.e.*, in a discrete-valued space. It is worth mentioning that the TCTC mechanism not only makes sense for our proposed MixSignGraph, but also works for other backbones. To verify this phenomenon, we also visualize the feature distribution of the other pre-trained backbone I3D in right part of Figure 4, which is adopted for How2Sign and OpenASL datasets (see Section 4 for details). The above phenomenon proves that TCTC can provide a weak tokenization function, even if TCTC may bring inaccurate segmentation.

## 4 Experiments

**Datasets.** Our experiment evaluation is conducted on five publicly available SL datasets, including **PHOENIX14** (25), **PHOENIX14T** (5),**CSL-Daily** (14), **How2sign** (26) and **OpenASL** (27). For more detailed descriptions of these datasets, please refer to Appendix A.2.

**Architecture Setting.** Our architecture is implemented using PyTorch 1.11. The setup includes the following components: (1) *Patchify stem and Patch merging*: We use a patchify stem and patch merging module in SignGraph to obtain patch (node) embeddings. (2) *Distance function*: In the baseline setting, we measure the distance between two nodes using the Euclidean distance. (3) *Global feature module*: This module comprises two 1D convolution blocks, a 2-layer BiLSTM with a hidden size of 1024 for global feature modeling, and a fully connected layer for the final prediction. (4) *Translation network*: For a fair comparison, following the current SOTA SLT model (7), we adopt the pre-trained mBART model provided by Hugging-face as our translation network.

Table 1: Effects of sign graph modules.

| $LSG_1$ | $TSG_1$ | $HSG_1$ | $LSG_2$ | $TSG_2$ | $HSG_2$ | Dev WER | Dev Del/Ins | Test WER | Test Del/ins |
|---|---|---|---|---|---|---|---|---|---|
| ✗ | ✗ | ✗ | ✗ | ✗ | ✗ | 22.3 | 8.4/2.5 | 22.2 | 8.1/2.7 |
| ✓ | | | | | | 19.2 | 5.6/2.2 | 21.0 | 4.8/2.3 |
| | ✓ | | | | | 19.6 | 5.6/2.1 | 21.5 | 5.1/2.5 |
| | | ✓ | | | | 19.9 | 6.7/2.1 | 21.6 | 6.2/3.1 |
| | | | ✓ | | | 19.3 | 5.3/2.1 | 20.8 | 5.8/2.0 |
| | | | | ✓ | | 19.5 | 6.6/1.7 | 21.2 | 5.4/2.4 |
| | | | | | ✓ | 19.4 | 6.2/2.9 | 21.5 | 5.9/3.6 |
| ✓ | | | ✓ | | | 18.7 | 5.1/2.3 | 20.6 | 5.2/1.7 |
| | ✓ | | | ✓ | | 18.6 | 4.3/1.8 | 20.2 | 5.5/1.7 |
| | | ✓ | | | ✓ | 17.8 | 6.1/2.3 | 19.6 | 5.3/2.9 |
| ✓ | ✓ | ✓ | | | | 17.1 | 5.3/2.0 | 19.7 | 5.0/3.0 |
| | | | ✓ | ✓ | ✓ | 17.4 | 5.7/2.1 | 19.6 | 5.1/2.9 |
| ✓ | ✓ | ✓ | ✓ | ✓ | ✓ | **16.7** | 4.9/2.1 | **19.0** | 4.9/2.9 |

**Training Setting.** For fair comparisons, we adopt the same data preprocessing and training settings as in previous work (2; 4). In regard to How2Sign and OpenASL datasets, we adopt the visual features generated by a pre-trained I3D model (8) . In addition, to ensure the model can be trained end-to-end on three 3090 GPUs with 24GB memory, the entire model is trained in half-precision.

**Evaluation.** We evaluate the performances of MixSignGraph on the following tasks: CSLR and SLT (including Sign2Gloss2Text, Sign2Text, Gloss-free Sign2Text). Please refer to Appendix A.3 for detailed descriptions of these tasks. To evaluate our model, we adopt the Word Error Rate (WER) metric for the CSLR task, and ROUGE-L F1 Score (28) and BLEU-1-4 (29) for SLT tasks.

### 4.1 Ablation Study

Following the previous work (5), we perform ablation studies on PHOENIX14T dataset to verify the effectiveness of the proposed MixSignGraph model.

**Effects of Sign Graph Modules.** As shown in Table 1, we conduct an ablation study to evaluate the individual and combined contributions of the LSG, TSG, and HSG modules. The baseline model without any graph module yields a WER of 22.3%/ 22.2% on the Dev/Test set. Furthermore, when HSG is combined with LSG and TSG, we observe consistent performance gains, which demonstrates that the proposed HSG module effectively mitigates the degradation of local features caused by cross-region feature extraction, and helps enhance the model's overall representation capability and CSLR performance.

Most importantly, the best performance is achieved when the HSG modules is inserted after LSG and TSG modules, resulting in the lowest WER of 16.7% (Dev) and 19.0% (Test), which demonstrates the importance of hierarchical local feature enhancement via HSG in complementing cross-region

Table 2: Effect of proposed TCTC for SLT. Besides, we also show the CSLR performance based on the pseudo gloss sequence obtained by text labels in the right part.

| Dataset | Model | Dev | | | | | Test | | | | | Dev | | Test | |
|---|---|---|---|---|---|---|---|---|---|---|---|---|---|---|---|
| | | ROUGE | BLEU1 | BLEU2 | BLEU3 | BLEU4 | ROUGE | BLEU1 | BLEU2 | BLEU3 | BLEU4 | WER | Del/Ins | WER | Del/Ins |
| PHOENIX14T | w/o TCTC | 32.63 | 33.57 | 18.71 | 13.47 | 9.20 | 34.56 | 35.31 | 21.05 | 16.45 | 9.40 | - | -/- | - | -/- |
| | w/ TCTC | 51.71 | 51.07 | 37.97 | 29.98 | 24.87 | 51.14 | 50.01 | 38.04 | 29.95 | 24.02 | 59.97 | 36.8/2.5 | 59.55 | 35.9/2.7 |
| | w/ gloss | 55.77 | 55.01 | 42.64 | 34.94 | 29.00 | 53.84 | 54.90 | 42.53 | 34.50 | 28.97 | 16.72 | 4.9/2.1 | 19.01 | 4.9/2.9 |
| CSL-Daily | w/o TCTC | 34.11 | 33.78 | 20.13 | 12.61 | 8.01 | 32.21 | 32.86 | 18.34 | 10.67 | 6.78 | | | | |
| | w/ TCTC | 49.16 | 49.98 | 36.42 | 26.89 | 20.43 | 49.93 | 50.24 | 36.91 | 27.54 | 21.01 | 66.87 | 39.11/4.9 | 66.05 | 38.18/4.9 |
| | w/ gloss | 54.54 | 55.87 | 42.45 | 32.75 | 25.77 | 54.67 | 55.41 | 42.43 | 32.84 | 25.87 | 25.13 | 6.4/2.1 | 25.01 | 7.0/1.6 |
| How2Sign | w/o TCTC | 18.35 | 23.88 | 12.92 | 7.89 | 5.05 | 18.06 | 23.41 | 12.55 | 7.61 | 4.86 | - | -/- | - | -/- |
| | w/ TCTC | 29.24 | 34.82 | 22.47 | 15.61 | 11.28 | 28.01 | 32.74 | 20.83 | 14.41 | 10.41 | 72.38 | 46.35/2.69 | 74.88 | 50.27/2.07 |
| OpenASL | w/o TCTC | 12.86 | 11.95 | 4.89 | 2.78 | 1.88 | 12.56 | 11.29 | 4.64 | 2.77 | 1.95 | - | -/- | - | -/- |
| | w/ TCTC | 25.41 | 26.82 | 16.70 | 11.48 | 8.36 | 25.71 | 26.65 | 16.55 | 11.68 | 8.69 | 81.53 | 59.88/1.08 | 81.33 | 60.39/1.29 |

Table 3: CSLR qualitative results on PHOENIX14, PHOENIX14T and CSL-Daily. We use different colors to represent substitutions, deletions, and insertions, respectively.

| example(a) | PHOENIX14 dataset |
|---|---|
| Groundtruth | MORGEN DASSELBE SCHAUER REGION SONST VIEL SONNE REGION TEILWEISE WEHEN STARK |
| MultiSignGraph | MORGEN DASSELBE SCHAUER REGION SONST VIEL SONNE REGION TEILWEISE WEHEN SCHWACH |
| MixSignGraph | MORGEN DASSELBE SCHAUER REGION SONST VIEL SONNE REGION TEILWEISE WEHEN STARK |
| example(b) | PHOENIX14T dataset |
| Groundtruth | DARUNTER NEBEL LANG IN-KOMMEND DANEBEN SONNE BERG OBEN DANN DURCHGEHEND SONNE |
| MultiSignGraph | __ON__ DARUNTER NEBEL LANG IN-KOMMEND DANEBEN SONNE BERG OBEN DANN DURCHGEHEND SONNE |
| MixSignGraph | DARUNTER NEBEL LANG IN-KOMMEND DANEBEN SONNE BERG OBEN DANN DURCHGEHEND SONNE |
| example(b) | CSL-Daily dataset |
| Groundtruth | 他 分别 年 十 学生 见面 |
| MultiSignGraph | 他 分别 年 十 同学 没有 见面 |
| MixSignGraph | 他 分别 年 十 同学 见面 |

Table 4: SLT qualitative results on PHOENIX14T, CSL-Daily, How2Sign and OpenASL.

| example(a) | PHOENIX14T dataset |
|---|---|
| Groundtruth | und zum wochenende wird es dann sogar wieder ein bisschen kälter |
| w/o TCTC | und zum wochenende teil recht freundliche hochdruckwetter begleitet uns |
| w/ TCTC | und zum wochenende wird es dann sogar ein bisschen kälter |
| Gloss-based | und zum wochenende wird es dann sogar wieder ein bisschen kälter |
| example(b) | CSL-daily dataset |
| Groundtruth | 这 件 事 分 别 有 什 么 好 处 和 坏 处？ |
| w/o TCTC | 这 事 有 什 么 好 处 和 坏 处? |
| w/ TCTC | 这 事 情 分 别 有 什 么 好 处 和 坏 处? |
| Gloss-based | 这 件 事 分 别 有 什 么 好 处 和 坏 处？ |
| example(c) | How2sign dataset |
| Groundtruth | You can take this forward and back, you can take it in a circle, you can take it in a lot of different directions. |
| w/o TCTC | Take a step forward, back; you can do a circular mothion that can go from on direction to the other. |
| w/ TCTC | You can take this forward and back, you can take it in circles, you can take it in a lot of different ways. |
| example(d) | OpenASL dataset |
| Groundtruth | There are results pending for 20 other tests |
| w/o TCTC | There are now fires and reports other people injured |
| w/ TCTC | There are waiting for the results of the 20 other tests |

spatial and temporal modeling. It confirms that $HSG$ plays a critical role in reinforcing local features and integrating multi-granularity graph-based representations.

**Effect of TCTC for Gloss-Free SLT.** To demonstrate the effectiveness of our proposed TCTC mechanism, we compare a SLT model trained end-to-end (*i.e.*, NOT using TCTC) and the same model pre-trained with TCTC, in terms of SLT performance. As shown in Table 2, our model pre-trained with TCTC significantly outperforms the one trained end-to-end, *e.g.*, our model improves gloss-free SLT performance by 23.14 ROUGE score and 19.8 BLEU4 score on PHOENIX14T dev set. In addition, on PHOENIX14T and CSL-Daily, the performance of our model pre-trained with TCTC is close to that of the gloss-based SLT model (*i.e.*, pre-trained with gloss annotations), *e.g.*, 55.77 ROUGE and 54.54 ROUGE score. It indicates that the proposed TCTC mechanism can substantially bridge the performance gap between GFSLT and gloss-based SLT.

## 4.2 Qualitative Results

**CSLR Qualitative Results.** As shown in Table 3, we conduct qualitative analysis for MixSignGraph in the CSLR task, and show one sample from the test set of PHOENIX14, PHOENIX14T and

Table 5: Comparison of CSLR performance on PHOENIX14 and PHOENIX14T datasets. (F: face, M: mouth, H: hands, S: skeleton, P: pre-training backbone with ImageNet, $\checkmark^*$: pre-training on other datasets. Same applies to the tables below.)

| Model | Backbone | Extra cues | | | | PHOENIX14 | | | | PHOENIX14T | |
| | | F/M | H | S | P | DEV | | TEST | | DEV | TEST |
| | | | | | | WER | del/ins | WER | del/ins | WER | WER |
|---|---|---|---|---|---|---|---|---|---|---|---|
| STMC (11) | VGG11 | ✓ | ✓ | ✓ | | 21.1 | 7.7/3.4 | 20.7 | 7.4/2.6 | 19.6 | 21.0 |
| C2SLR (30) | ResNet18 | | | ✓ | | 20.5 | -/- | 20.4 | -/- | 20.2 | 20.4 |
| TwoStream (7) | S3D | ✓ | ✓ | ✓ | ✓ | 18.4 | -/- | 18.8 | -/- | 17.7 | 19.3 |
| CrossL-Two (3) | S3D | ✓ | ✓ | ✓ | ✓* | 15.7 | -/- | 16.7 | -/- | 16.9 | 18.5 |
| CrossL-Single (3) | S3D | | | | ✓* | - | -/- | - | -/- | 20.6 | 21.3 |
| RTG-Net (12) | RepVGG | ✓ | ✓ | ✓ | ✓ | 20.0 | 8.4/1.5 | 20.1 | 8.6/1.7 | 19.63 | 20.01 |
| Joint-SLRT (31) | GooleNet | | | | ✓ | - | - | - | - | 24.6 | 24.5 |
| TwoStream (7) | S3D | | | | ✓* | 22.4 | -/- | 23.3 | -/- | 21.1 | 22.4 |
| VAC (2) | ResNet18 | | | | ✓ | 21.2 | 7.9/2.5 | 22.3 | 8.4/2.6 | - | - |
| SMKD (15) | ResNet18 | | | | ✓ | 20.8 | 6.8/2.5 | 21.0 | 6.3/2.3 | 20.8 | 22.4 |
| CorrNet (32) | ResNet18 | | | | ✓ | 18.8 | 5.6/2.8 | 19.4 | 5.7/2.3 | 18.9 | 20.5 |
| FCN (33) | Customed | | | | | 23.7 | -/- | 23.9 | -/- | - | - |
| Contrastive (6) | ResNet18 | | | | ✓ | 19.6 | 5.1/2.7 | 19.8 | 5.8/3.0 | 20.0 | 20.1 |
| HST-GNN (34) | Customized(GCN) | ✓ | ✓ | ✓ | ✓ | 19.5 | -/- | 19.8 | -/- | 19.5 | 19.8 |
| CoSign (35) | ST-GCN(GCN) | ✓ | ✓ | ✓ | | 19.7 | -/- | 20.1 | -/- | 19.5 | 20.1 |
| SignGraph (4) | Customized(GCN) | | | | ✓ | 18.2 | 4.9/2.0 | 19.1 | 5.3/1.9 | 17.8 | 19.1 |
| MixSignGraph | Customized(GCN) | | | | ✓ | 16.5 | 4.9/2.0 | 17.3 | 4.9/2.2 | 16.7 | 19.0 |

CSL-Daily, respectively. It can be found that MixSignGraph yields more accurate gloss predictions than MultiSignGraph, demonstrating that the newly-proposed MixSignGraph is more effective.

**SLT Qualitative Results.** As shown in Table 4, we present qualitative analysis of MixSignGraph in the SLT task, and show one sample from the test set of PHOENIX14T, CSL-Daily, How2Sign and OpenASL, respectively. It shows that the gloss-based Sign2Text model achieves the best performance while MixSignGraph pre-trained with TCTC also achieves satisfactory translation results.

## 4.3 Comparisons on CSLR Tasks

**Evaluation on PHOENIX14T.**
As shown in Table 5, we compare our model with existing models on CSLR performance, and we provide both the performances on the validation set and test set. Most of the current models adopt existing CNN-based backbones, and achieve good performance by injecting extra cues (11; 7; 30), adding extra constraints (6; 2; 15) or introducing attention mechanism (32). As for the GCN-based models, CoSign (35) mainly relies on pre-processed fine-grained skeleton data, and achieves 19.5%, 20.1% WER on dev, test set. While HST-GCN (34) adopts both CNN-based backbone and GCN-based backbone to extract RGB features and skeleton features respectively, and achieves 19.5%, 19.8% WER on dev, test set. It is worth mentioning that the SOTA model CrossL-Two (3) utilizes both RGB features and fine-grained skeleton features (*i.e.*, keypoints in hands, body and face), and pre-trains its backbone on both PHOENIX14 and CSL-daily datasets, achieving 16.9%, 18.5% WER on dev, test set respectively. When moving to our MixSignGraph, it does not use any extra cues or pre-train on other SL datasets, but it still achieves comparable performance with CrossL-Two on test set. While on dev set, our model even outperforms CrossL-Two model by 0.2% WER.

Table 6: Comparison of CSLR performance on CSL-daily.

| CSLR | Backbone | Extra cues | | DEV | | TEST | |
| | | S | P | WER | del/ins | WER | del/ins |
|---|---|---|---|---|---|---|---|
| Joint-SLRT (31) | GoogleNet | | ✓ | 33.1 | 10.3/4.4 | 32.0 | 9.6/4.1 |
| TwoStream (7) | S3D | ✓ | ✓* | 25.4 | -/- | 25.3 | -/- |
| TwoStream (7) | S3D | | ✓* | 28.9 | -/- | 28.5 | -/- |
| BN-TIN (14) | GoogLeNet | | ✓ | 33.6 | 13.9/3.4 | 33.1 | 13.5/3.0 |
| CorrNet (32) | ResNet18 | | ✓ | 30.6 | -/- | 30.1 | -/- |
| Contrastive (6) | ResNet18 | | ✓ | 26.0 | 11.5/3.0 | 25.3 | 11.2/3.5 |
| CoSign (35) | ST-GCN | ✓ | | 28.1 | -/- | 27.2 | -/- |
| SignGraph (4) | GCN | | ✓ | 27.3 | 7.9/2.3 | 26.4 | 7.8/2.1 |
| MixSignGraph | GCN | | ✓ | 25.1 | 6.4/2.1 | 25.0 | 7.0/1.6 |

**Evaluation on PHOENIX14.** We conduct a comparative analysis of our model and current CSLR models using the PHOENIX14 dataset. As illustrated in Table 5, our straightforward yet robust MixSignGraph model achieves the superior performance (16.5% WER) on the dev set and surpasses the majority of existing models, thus demonstrating the effectiveness of our MixSignGraph.

**Evaluation on CSL-daily.** We also show CSLR performance of our model on CSL-daily dataset. Table 6 shows that our model achieves 25.1%, 25.0% WER on the dev, test set, respectively. It indicates that our model can surpass the SOTA model by only using RGB modality.

Table 7: Comparison of SLT performance on PHOENIX14T dataset.

| Sign2Gloss2Text | Extra cues | | | | PHOENIX14T DEV | | | | | TEST | | | | |
|---|---|---|---|---|---|---|---|---|---|---|---|---|---|---|
| | F/M | H | S | P | ROUGE | BLEU1 | BLEU2 | BLEU3 | BLEU4 | ROUGE | BLEU1 | BLEU2 | BLEU3 | BLEU4 |
| SL-Luong (5) | | | | ✓ | 44.14 | 42.88 | 30.30 | 23.02 | 18.40 | 43.80 | 43.29 | 30.39 | 22.82 | 18.13 |
| Joint-SLRT (31) | | | | ✓ | 47.73 | 34.82 | 27.11 | 22.11 | - | 48.47 | 35.35 | 27.57 | 22.45 | |
| SignBT (14) | | | | ✓ | 49.53 | 49.33 | 36.43 | 28.66 | 23.51 | 49.35 | 48.55 | 36.13 | 28.47 | 23.51 |
| STMC-Transf (36) | ✓ | ✓ | ✓ | ✓ | 46.31 | 48.27 | 35.20 | 27.47 | 22.47 | 46.77 | 48.73 | 36.53 | 29.03 | 24.00 |
| MMTLB (22) | | | | ✓ | 50.23 | 50.36 | 37.50 | 29.69 | 24.63 | 49.59 | 49.94 | 37.28 | 29.67 | 24.60 |
| RTG-Net (12) | ✓ | ✓ | ✓ | ✓ | 50.18 | 51.17 | 37.95 | 29.88 | 25.95 | 50.04 | 50.87 | 37.95 | 29.74 | 25.87 |
| TwoStream-SLT (7) | ✓ | ✓ | ✓ | ✓ | 52.01 | 52.35 | 39.76 | 31.85 | 26.47 | 51.59 | 52.11 | 39.81 | 32.00 | 26.71 |
| MixSignGraph | | | | ✓ | 52.59 | 52.40 | 39.84 | 31.95 | 26.56 | 51.46 | 52.35 | 39.23 | 32.26 | 26.04 |
| Sign2Text | F/M | H | S | P | ROUGE | BLEU1 | BLEU2 | BLEU3 | BLEU4 | ROUGE | BLEU1 | BLEU2 | BLEU3 | BLEU4 |
| Joint-SLRT (31) | | | | ✓ | - | 47.26 | 34.40 | 27.05 | 22.38 | - | 46.61 | 33.73 | 26.19 | 21.32 |
| HST-GNN | ✓ | ✓ | ✓ | | | 46.10 | 33.40 | 27.50 | 22.6 | | 45.20 | 34.70 | 27.50 | 22.60 |
| STMC-T (10) | ✓ | ✓ | ✓ | ✓ | 48.24 | 47.60 | 36.43 | 29.18 | 24.09 | 46.65 | 46.98 | 36.09 | 28.70 | 23.65 |
| SignBT (14) | | | | ✓ | 50.29 | 51.11 | 37.90 | 29.80 | 24.45 | 49.54 | 50.80 | 37.75 | 29.72 | 24.32 |
| CoSLRT (6) | | | | | 52.47 | 52.29 | 39.60 | 31.34 | 27.83 | 52.24 | 52.48 | 41.17 | 32.30 | 27.88 |
| MMTLB (22) | | | | ✓ | 53.10 | 53.95 | 41.12 | 33.14 | 27.61 | 52.65 | 53.97 | 41.75 | 33.84 | 28.39 |
| TwoStream-SLT (7) | ✓ | ✓ | ✓ | ✓ | 54.08 | 54.32 | 41.99 | 34.15 | 28.66 | 53.48 | 54.90 | 42.43 | 34.46 | 28.95 |
| MixSignGraph | | | | ✓ | 55.77 | 55.01 | 42.64 | 34.94 | 29.00 | 53.84 | 54.90 | 42.53 | 34.50 | 28.97 |
| Gloss-free SLT | F/M | H | S | P | ROUGE | BLEU1 | BLEU2 | BLEU3 | BLEU4 | ROUGE | BLEU1 | BLEU2 | BLEU3 | BLEU4 |
| SL-Luong (5) | | | | ✓ | 31.80 | 31.87 | 19.11 | 13.16 | 9.94 | 31.80 | 32.24 | 19.03 | 12.83 | 9.58 |
| TSPNet (37) | | | | ✓ | - | - | - | - | - | 34.96 | 36.10 | 23.12 | 16.88 | 13.41 |
| GASLT (16) | | | | | - | - | - | - | - | 39.86 | 39.07 | 26.74 | 21.86 | 15.74 |
| CSGCR (38) | | | | ✓ | 38.96 | 35.85 | 24.77 | 18.65 | 15.08 | 38.85 | 36.71 | 25.40 | 18.86 | 15.18 |
| GFSLT (16) | | | | ✓ | 40.93 | 41.97 | 31.04 | 24.30 | 19.84 | 40.70 | 41.39 | 31.00 | 24.20 | 19.66 |
| GFSLT-VLP (16) | | | | ✓ | 43.72 | 44.08 | 33.56 | 26.74 | 22.12 | 42.49 | 43.71 | 33.18 | 26.11 | 21.44 |
| Sign2GPT (39) | | | | ✓* | - | - | - | - | - | 48.90 | 49.54 | 35.96 | 28.83 | 22.52 |
| MixSignGraph | | | | ✓ | 51.71 | 51.07 | 37.97 | 29.98 | 24.87 | 51.14 | 50.01 | 38.04 | 29.95 | 24.02 |

Table 8: Comparison of SLT performance on CSL-Daily dataset.

| Sign2Gloss2Text | Extra cues | | | | CSL-Daily DEV | | | | | TEST | | | | |
|---|---|---|---|---|---|---|---|---|---|---|---|---|---|---|
| | F/M | H | S | P | ROUGE | BLEU1 | BLEU2 | BLEU3 | BLEU4 | ROUGE | BLEU1 | BLEU2 | BLEU3 | BLEU4 |
| SL-Luong (5) | | | | ✓ | 40.18 | 41.46 | 25.71 | 16.57 | 11.06 | 40.05 | 41.55 | 25.73 | 16.54 | 11.03 |
| Joint-SLRT (31) | | | | ✓ | 44.18 | 46.82 | 32.22 | 22.49 | 15.94 | 44.81 | 47.09 | 32.49 | 22.61 | 16.24 |
| SignBT (14) | | | | ✓ | 48.38 | 50.97 | 36.16 | 26.26 | 19.53 | 48.21 | 50.68 | 36.00 | 26.20 | 19.67 |
| MMTLB (22) | | | | ✓ | 51.35 | 50.89 | 37.96 | 28.53 | 21.88 | 51.43 | 50.33 | 37.44 | 28.08 | 21.46 |
| TwoStream-SLT (7) | ✓ | ✓ | ✓ | ✓ | 53.91 | 53.58 | 40.49 | 30.67 | 23.71 | 54.92 | 54.08 | 41.02 | 31.18 | 24.13 |
| MixSignGraph | | | | ✓ | 53.48 | 53.73 | 41.53 | 30.70 | 23.76 | 53.65 | 54.07 | 41.04 | 31.32 | 24.44 |
| Sign2Text | F/M | H | S | P | ROUGE | BLEU1 | BLEU2 | BLEU3 | BLEU4 | ROUGE | BLEU1 | BLEU2 | BLEU3 | BLEU4 |
| Joint-SLRT (31) | | | | ✓ | 37.06 | 37.47 | 24.67 | 16.86 | 11.88 | 36.74 | 37.38 | 24.36 | 16.55 | 11.79 |
| SignBT (14) | | | | ✓ | 49.49 | 51.46 | 37.23 | 27.51 | 20.80 | 49.31 | 51.42 | 37.26 | 27.76 | 21.34 |
| Contrastive (6) | | | | | 50.34 | 51.97 | 37.10 | 27.53 | 21.79 | 50.73 | 52.31 | 37.37 | 27.89 | 21.81 |
| MMTLB (22) | | | | ✓ | 53.38 | 53.81 | 40.84 | 31.29 | 24.42 | 53.25 | 53.31 | 40.41 | 30.87 | 23.92 |
| TwoStream-SLT (7) | ✓ | ✓ | ✓ | ✓ | 55.10 | 55.21 | 42.31 | 32.71 | 25.76 | 55.72 | 55.44 | 42.59 | 32.87 | 25.79 |
| MixSignGraph | | | | ✓ | 54.54 | 55.87 | 42.45 | 32.75 | 25.77 | 54.67 | 55.41 | 42.43 | 32.84 | 25.87 |
| Gloss-free SLT | F/M | H | S | P | ROUGE | BLEU1 | BLEU2 | BLEU3 | BLEU4 | ROUGE | BLEU1 | BLEU2 | BLEU3 | BLEU4 |
| SL-Luong (5) | | | | ✓ | 34.28 | 34.22 | 19.72 | 12.24 | 7.96 | 34.54 | 34.16 | 19.57 | 11.84 | 7.56 |
| GASLT (16) | | | | ✓ | - | - | - | - | | 20.35 | 19.90 | 9.94 | 5.98 | 4.07 |
| GFSLT-VLP (17) | | | | ✓ | 36.44 | 39.20 | 25.02 | 16.35 | 11.07 | 36.70 | 39.37 | 24.93 | 16.26 | 11.00 |
| Sign2GPT (39) | | | | ✓* | - | - | - | - | - | 42.36 | 41.75 | 28.73 | 20.60 | 15.40 |
| GFSLT-VLP-SignCL (40) | | | | ✓ | - | - | - | - | - | 49.04 | 49.76 | 36.85 | 29.97 | 22.74 |
| SignLLM (18) | | | | ✓* | 44.49 | 46.88 | 36.59 | 29.91 | 25.25 | 47.23 | 45.21 | 34.78 | 28.05 | 23.40 |
| MixSignGraph | | | | ✓ | 49.16 | 49.98 | 36.42 | 26.89 | 20.43 | 49.93 | 50.24 | 36.91 | 27.54 | 20.78 |

Table 9: Comparison of SLT performance on How2Sign dataset.

| Gloss-free SLT | Extra cues | | | | How2Sign DEV | | | | | TEST | | | | |
|---|---|---|---|---|---|---|---|---|---|---|---|---|---|---|
| | F/M | H | S | P | ROUGE | BLEU1 | BLEU2 | BLEU3 | BLEU4 | ROUGE | BLEU1 | BLEU2 | BLEU3 | BLEU4 |
| SLT-IV (8) | | | | ✓ | - | 35.20 | 20.62 | 13.25 | 8.89 | - | 34.01 | 19.30 | 12.18 | 8.03 |
| YouTube-SLT (41) | | | | ✓ | - | - | - | - | - | - | 14.96 | 5.11 | 2.26 | 1.22 |
| SSVP-SLT (42) | | | | ✓ | - | - | - | - | - | 25.70 | 30.20 | 16.70 | 10.50 | 7.00 |
| GloFE-VN (43) | | | | ✓ | 12.98 | 15.21 | 7.38 | 4.07 | 2.37 | 12.61 | 14.94 | 7.27 | 3.93 | 2.24 |
| Ours(I3D) | | | | ✓ | 29.24 | 34.82 | 22.47 | 15.61 | 11.28 | 28.01 | 34.74 | 20.83 | 14.41 | 10.41 |

## 4.4 Comparisons on SLT tasks

**Evaluation on PHOENIX14T Dataset.** As shown in Table 7, we compare our model with existing models using PHOENIX14T dataset on SLT tasks, which include Sign2Gloss2Text, Sign2Text, and Gloss-free Sign2Text (also known as Gloss-free SLT). Our MixSignGraph achieves comparable performance to the SOTA model (TwoStream) on both Sign2Gloss2Text and Sign2Text tasks. When moving to the gloss-free SLT task, our MixSignGraph pre-trained with TCTC mechanism significantly surpasses the current SOTA model, *e.g.*, 8.65 ROUGE score higher than GFSLT-VLP (16) on test set. Besides, MixSignGraph also narrows the performance gap between the gloss-free SLT model and the gloss-based SLT model, *i.e.*, 51.14 ROUGE score vs. 53.84 ROUGE score.

**Evaluation on CSL-daily.** As shown in Table 8, we also show the SLT performance of our MixSign-Graph on CSL-daily dataset. It can be found that MixSignGraph outperforms the most of current

Table 10: Comparison of SLT performance on OpenASL dataset.

| Gloss-free SLT | Extra cues | | | | OpenASL DEV | | | | | TEST | | | | |
|---|---|---|---|---|---|---|---|---|---|---|---|---|---|---|
| | F/M | H | S | P | ROUGE | BLEU1 | BLEU2 | BLEU3 | BLEU4 | ROUGE | BLEU1 | BLEU2 | BLEU3 | BLEU4 |
| Conv-GRU(27) | | | | ✓ | 16.25 | 16.72 | 8.95 | 6.31 | 4.82 | 16.10 | 16.11 | 8.85 | 6.18 | 4.58 |
| I3D-transformer(27) | | | | ✓ | 18.88 | 18.26 | 10.26 | 7.17 | 5.60 | 18.64 | 18.31 | 10.15 | 7.19 | 5.66 |
| Open | ✓ | ✓ | | | 25.31 | 24.35 | 14.94 | 10.72 | 8.39 | 24.83 | 23.87 | 14.08 | 9.90 | 7.54 |
| Open(27) | ✓ | ✓ | | ✓ | 20.43 | 20.10 | 11.81 | 8.43 | 6.57 | 21.02 | 20.92 | 12.08 | 8.59 | 6.72 |
| GloFE-VN (43) | | | | ✓ | 21.37 | 21.06 | 12.34 | 8.68 | 6.68 | 21.75 | 21.56 | 12.74 | 9.05 | 7.06 |
| Ours(I3D) | | | | ✓ | 25.41 | 26.82 | 16.70 | 11.48 | 8.36 | 25.71 | 26.65 | 16.55 | 11.68 | 8.69 |

models on all the Sign2Gloss2Text, Sign2Text and gloss-free SLT tasks, especially on the gloss-free SLT task (*i.e.*, 2.7 ROUGE score higher than the SOTA model).

**Evaluation on How2Sign.** How2Sign is a large American Sign Language dataset. Considering that the How2Sign dataset only provides sentence labels (*i.e.*, NO gloss labels), we compare our proposed model with existing models in terms of gloss-free SLT performance. Note that we adopted the I3D features provided by previous work to accelerate training, which may compromise the SLT performance. As shown in Table 9, our MixSignGraph pre-trained with TCTC achieves a BLEU4 score of 11.28/10.41 on the dev/test set, outperforming the baseline model (which also uses I3D features) and thus establishing a new baseline for future work.

**Evaluation on OpenASL.** Similar to the How2Sign dataset, OpenASL is also a large dataset that contains only sentence labels and has a very rich vocabulary. We compare our proposed model with these baseline models, which adopt I3D as backbone for fair comparisons in terms of gloss-free SLT performance. As shown in Table 10, our model outperforms the baseline models and provides a new baseline for future work.

# 5   Conclusion

We propose MixSignGraph, a simple yet effective architecture that models sign sequences as graphs for feature extraction. To address local feature degradation caused by cross-region modeling in SignGraph, we design a Hierarchical Sign Graph (HSG) module that connects corresponding regions across multi-granularity features by bidirectional information exchange. We further introduce a Text-based CTC Pre-training (TCTC) strategy that generates pseudo-gloss sequences via simple NLP processing. Experiments on five benchmarks across CSLR and SLT tasks show that our model achieves strong performance without relying on any additional cues.

# 6   Limitations and Broader Impacts

Here, we list some potential ideas that can be further explored to improve the performance of SL tasks. Our TCTC mechanism for the gloss-free SLT task works well based on processed text, but it also encounters problems such as difficulty in CTC loss convergence, especially in cases with large vocabularies. Therefore, more effective methods to obtain pseudo labels for CTC pre-training are expected. In addition, we also highlight the potential negative social impacts. First, our method may experience unpredictable failures, so it should not be used in scenarios where such failures could have serious consequences. Second, our method is a data-driven approach and its performance may be influenced by biases in data, thus, caution is advised in the data collection process.

# 7   Acknowledgments

This work is supported in part by National Natural Science Foundation of China under Grant Nos. 62172208, 92467202, 62272216; Natural Science Foundation of Jiangsu Province (Key Program) under Grant No. BK20243040. This work is partially supported by Collaborative Innovation Center of Novel Software Technology and Industrialization. Training and testing were partially supported by Sulis Tier 2 HPC platform hosted by the Scientific Computing Research Technology Platform at the University of Warwick (EPSRC Grant EP/T022108/1 and the HPC Midlands+ consortium).

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

# A  Appendix

## A.1  Related Work

**Sign Language Tasks.**  The unique grammatical rules of sign language lead to some differences between the frameworks of CSLR and SLT tasks. Nevertheless, both frameworks contain a visual backbone to capture visual features from videos. However, it is a challenging task for the widely adopted CNN-based backbones to focus on sign-related features, since CNN backbones are initially designed to capture the texture and contours of objects (44). To guide CNN-based backbones to extract sign-related features, some current SL models introduce prior knowledge (*e.g.*, local areas (45), skeletons (9; 30; 7) and depth images (13)) to manually inject expert knowledge into the backbone, making it focus on SL-related areas. Some other SL models try to add additional constraints (*e.g.*, knowledge distillation (15; 2) and contrastive learning (6)) to optimize their backbones. Besides, there are also some SL models that turn their attention to obtaining more sign language samples by back translation (14), cross modality augmentation (46), pre-training with other sign language related datasets (22; 3) and adopting pre-trained visual models as their backbones (39). To capture effective SL-related features, SignGraph (4) represents frames as nodes and builds LSG and TSG modules to capture cross-region features both in one frame (*i.e.*, the correlation of hands and face regions) and among adjacent frames (*e.g.*, hand motions, facial expression changes).

In CSLR tasks, the CTC module (47) is adopted to align gloss sequences with SL videos. While SLT (5) models often adopt a pre-trained language model (*e.g.*, mBART (23), GPT2 (48)) to get the translated sentence. In these SLT models, the traditional training paradigm usually involves two stages: first, pre-training their backbone using gloss sequences (equivalent to the CSLR task), then connecting a pre-trained translation model and fine-tuning the entire model. In the first stage, labeled gloss sequence is critical for improving SLT performances, as pre-training with gloss sequences can assist the model in learning segmentation and semantic information of SL. However, annotating gloss sequences for SL videos is labor-intensive and cumbersome. Therefore, gloss-free SLT tasks have gained attention (16; 17; 39; 40). The existing gloss-free SLT models usually adopt attention mechanisms (16) or contrastive language-image pre-training (17) to improve SLT performances, and have achieved promising results.

**Graph Convolutional Network.**  Due to the graph structures, graph convolution layers (49) are more flexible for message passing and aggregation. Naturally, GCN has been effectively applied to skeleton-based data, where the connections between joints render the skeleton data inherently suitable for GCN. With the skeleton data, GCN has been applied to SL tasks, in which some models utilized GCNs to guide CNN-based backbones to extract skeleton-related features (9; 50; 34; 35). However, these approaches often rely heavily on structured inputs (*e.g.*, skeleton). To reduce the dependence on structured data, the recent model VIT (51), which processes an image as patches with a transformer model, has replaced CNNs as a de facto architecture in many fields. After that, the VIG model (19) directly represents an image as a graph structure to capture irregular and complex objects, and achieves superior performance on image recognition tasks.

## A.2  Datasets

Our experiment evaluation is conducted on five publicly available SL datasets, including three widely used datasets (*i.e.*, PHOENIX, PHOENIX14T and CSL-Daily) and two new and large datasets (*i.e.*, How2Sign and OpenASL), as described below.

- **PHOENIX14** (25): A widely used German SL dataset with 1295 glosses from 9 signers for CSLR. It includes 5672, 540 and 629 weather forecast samples for training, validation, and testing, respectively.
- **PHOENIX14T** (5): A German SL dataset with both gloss annotations and translation annotations. It contains 7096, 519 and 642 samples from 9 signers for training, validation and testing, respectively. In regard to the two-stage annotations, the sign gloss annotations have a vocabulary of 1066 different signs for CSLR, while the German translation annotations have a vocabulary of 2877 different words for SLT.
- **CSL-Daily** (14): A Chinese SL dataset with 18401, 1176, 1077 labeled videos from 10 signers for training, testing and validation. It includes 2000 gloss annotations for CSLR and 2343 Chinese words for SLT.

| Dataset | Video Samples | | | Gloss Vocabulary | | | Token Vocabulary | | | Vocabulary in TCTC | | |
|---|---|---|---|---|---|---|---|---|---|---|---|---|
| | Train | Test | Validation | Train | Test | Validation | Train | Test | Validation | Train | Test | Validation |
| PHOENIX14 (25) | 5672 | 629 | 540 | 1103 | 497 | 462 | - | - | - | - | - | - |
| PHOENIX14T (5) | 7096 | 642 | 519 | 1085 | 411 | 393 | 2143 | 976 | 927 | 2888 | 1002 | 952 |
| CSL-Daily (14) | 18401 | 1176 | 1077 | 2000 | 1345 | 1358 | 2342 | 1358 | 1344 | 2332 | 1351 | 1351 |
| How2Sign (26) | 30904 | 2328 | 1713 | - | - | - | 8816 | 3312 | 3046 | 9098 | 2573 | 2302 |
| OpenASL (27) | 96476 | 975 | 966 | - | - | - | 13902 | 3336 | 3187 | 19144 | 2362 | 2220 |

Table 11: Details of datasets used in our paper.

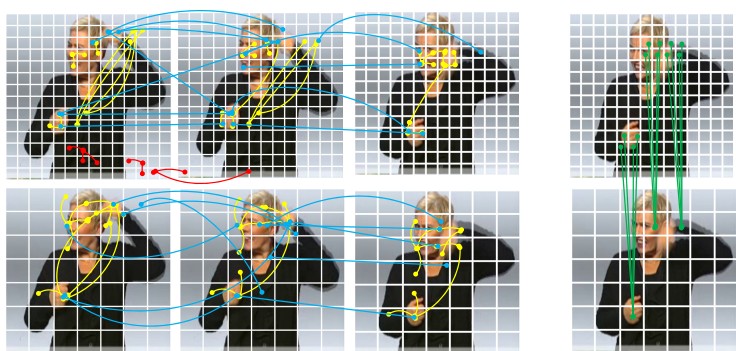

Figure 5: Visualization of graph construction: $LSG$ and $TSG$ modules ($LSG_1$ and $TSG_1$ in the first row, $LSG_2$ and $TSG_2$ in the second row) are in the left part, $HSG$ module is in the right part. The graph in $LSG$, $TSG$ and $HSG$ module is shown in yellow, blue and green, respectively. We also show some 'unimportant' edges between nodes of the background in red color.

- **How2Sign** (26): A multimodal and multiview continuous American Sign Language (ASL) dataset. It consists of a parallel corpus, which has more than 80 hours of sign language videos and a set of corresponding modalities (*i.e.*, speech, English transcripts, and depth). In this dataset, there are 31128, 1741 and 2322 samples for training, validation and testing, respectively. In regard to the annotations, there are no gloss-level annotations, thus the dataset is used for SLT tasks. It is worth mentioning that our experiment only uses the RGB modality (*i.e.*, NOT all modalities) with frontal view (*i.e.*, NOT all views) to train and test our model.

- **OpenASL** (27): A large ASL dataset with translation annotations for SLT. It includes 280 hours of ASL videos from more than 200 signers, and contains 96476, 997 and 999 samples for training, validation and testing, respectively.

We provide detailed information of the five datasets in Table 11. In addition, we also provide the token vocabulary obtained by the mBART tokenizer and the vocabulary of pseudo gloss labels in TCTC for reference. It is worth noting that the datasets we downloaded (*i.e.*, shown in Table 11) may slightly differ from the official versions described in their own papers.

## A.3 Evaluation Tasks.

We evaluate the performance of our model on the following tasks: : CSLR and SLT (including Sign2Gloss2Text, Sign2Text, Gloss-free Sign2Text. (1) **CSLR:** Recognizing a sign sequence as a corresponding gloss sequence, also known as Sign2Gloss. (2) **SLT**: Translating a sign sequence into a spoken sentence, and it can be classified into the following three categories: (2.1) **Sign2Gloss2Text:** First, the recognized gloss sequence is obtained based on the CSLR model. Then, the predicted glosses are translated to a spoken sentence by a translation model. (2.2) **Sign2Text**: Directly translating a SL video into a spoken sentence. Gloss annotations are required for pre-training a CSLR model, and visual features output by the CSLR model are sent to a translation model for generating the spoken sentence. (2.3) **Gloss-free Sign2Text:** Directly translating a SL video into a spoken sentence, while NOT using gloss annotations to pretrain a CSLR model. This task is also known as gloss-free SLT.

Table 12: Model details.

| Model | Task | Visual backbone | Translation model | FLOPs (G) | Parameters(M) | Pre-trained Dataset |
|---|---|---|---|---|---|---|
| CrossL-Two | CSLR | Dual S3D | - | - | - | CSL-Daily+Phoenix14+Phoenix14T |
| Sign2GPT | Gloss-free SLT | Dino-V2 ViT | GPT2 | - | 1771.65 | LVD-142M+WebText |
| TwoStream | CSLR+S2T+S2G2T | Dual S3D | mBART | 323.41 | 405.38 | Kinetics-400+CC25 |
| MultiSigngraph | CSLR+S2T+S2G2T+gloss-free SLT | SignGraph | mBART | 256.13 | 395.80 | ImageNet-1k+CC25 |
| MixSigngraph | CSLR+S2T+S2G2T+gloss-free SLT | SignGraph | mBART | 320.88 | 402.20 | ImageNet-1k+CC25 |

### A.4 Visualization of MixSignGraph

To verify whether our model can effectively capture sign-related features, we select a sign video from the PHOENIX14T test set and visualize the constructed graph structure in both stages. In Figure 5, we show the graphs in the $LSG$, $TSG$ and $HSG$ modules. For clarity, only a subset of the edges is displayed. In the $LSG$ module, our model links nodes with similar contents or related semantic representation (*e.g.*, hand regions and face regions), to extract better intra-frame cross-region features. In the $TSG$ module, our model builds edges among adjacent frames to track dynamic changes in gestures and facial expressions, to capture inter-frame cross-region features. In the $HSG$ module, our model connects the same regions with different granularities (*i.e.*, the corresponding regions in feature maps), to enhance one-region features. In addition, we also highlight some 'unimportant' edges between nodes in the background in red. As seen in Figure 5, the background nodes in the $LSG$ module are naturally connected to their neighboring nodes, and there are still a few background nodes connected in the $TSG$ module. Fortunately, background nodes do not 'disturb' nodes in sign-related regions, demonstrating the effectiveness of our model.

### A.5 Model Comparison and Analysis

To provide a more comprehensive analysis, we compare our model with the existing models in the following aspects: tasks, architecture information, flops, number of parameters, and pre-trained datasets. As shown in Table 12, among the existing models, the TwoStream model adopts a dual S3D backbone and pre-train their backbone with Kinetics-400, which is an action recognition dataset. Based on the backbone in TwoStream, the CrossL-Two model further pre-trains their backbone with existing SL datasets. Sign2GPT adopts Dino-V2 ViT as the visual backbone, which is pre-trained on LVD-142M (a very large vision dataset), and adopts GPT2 as the translation model, which is also pre-trained on a very large text dataset WebText. In regard to MixSignGraph, it contains fewer parameters and FLOPs, and it is only pre-trained on small datasets (*i.e.*, ImageNet-1K and CC25 datasets), while achieving the promising performance on multiple SL tasks.

### A.6 Other Ablation Study

**Effect of Hyperparameter** $\mathsf{K}$**.** In the MixSignGraph model, four hyperparameters: $\mathsf{K}_l^1$ and $\mathsf{K}_l^2$ in the $LSG$ modules, $\mathsf{K}_t^1$ and $\mathsf{K}_t^2$ in the $TSG$ modules need to be set to proper values. To determine the best values for these hyperparameters, we evaluate the CSLR performance by changing $\mathsf{K}_l$ (*i.e.*, both $\mathsf{K}_l^1$ and $\mathsf{K}_l^2$), $\mathsf{K}_t$ (*i.e.*, both $\mathsf{K}_t^1$ and $\mathsf{K}_t^2$) values from 2 to 9, 7 to 91, respectively. However, due to the large number of possible combinations of $\mathsf{K}_l^1$, $\mathsf{K}_l^2$, $\mathsf{K}_t^1$ and $\mathsf{K}_t^2$, it is impractical to make an exhaustive search to find the globally optimal values of

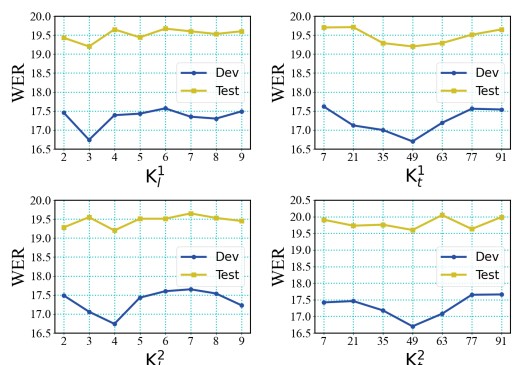

Figure 6: Effects of $\mathsf{K}_l$ and $\mathsf{K}_t$.

these hyperparameters. Therefore, we fix three of the parameters at a time and adjust the fourth to find suitable values. Finally, based on the WER performance on the DEV set, we set $\mathsf{K}_l^1$, $\mathsf{K}_l^2$, $\mathsf{K}_t^1$ and $\mathsf{K}_t^2$ to 3, 4, 49 and 49, respectively, as shown in Figure 6.

**Effect of Graph Type.** In MixSignGraph, we build dynamic dense graphs (*i.e.*, connecting the top $\mathsf{K}$ neighbors for each node) in the $LSG$ module, dynamic sparse graphs (*i.e.*, connecting only the top $\mathsf{K}$

# Table 13: Effect of porposed graph modules

### (a) Effect of graph types.

| $LSG_1$ | $TSG_1$ | $BSG_1$ | Dev WER | Dev Del/Ins | Test WER | Test Del/ins |
|---|---|---|---|---|---|---|
| Dense | Sparse | Fixed | **16.7** | 4.9/2.1 | **19.0** | 4.9/2.9 |
| Dense | Dense | Fixed | 19.3 | 4.9/2.9 | 20.1 | 5.2/2.6 |
| Sparse | Sparse | Fixed | 19.0 | 5.3/2.1 | 20.2 | 5.4/2.1 |
| Sparse | Dense | Fixed | 20.2 | 5.3/2.6 | 20.6 | 5.5/2.9 |
| Dense | Sparse | Dynamic | 18.1 | 5.1/2.2 | 20.1 | 5.1/2.1 |
| Dense | Dense | Dynamic | 19.9 | 5.6/3.5 | 21.8 | 5.5/2.6 |
| Sparse | Sparse | Dynamic | 19.1 | 5.0/1.9 | 21.1 | 5.8/2.1 |
| Sparse | Dense | Dynamic | 20.7 | 6.0/2.1 | 21.5 | 6.0/2.5 |

### (b) Effect of order of proposed sign graph modules.

| Model | Dev WER | Dev Del/Ins | Test WER | Test Del/ins |
|---|---|---|---|---|
| HSG $\rightarrow$ LSG $\rightarrow$ TSG | 16.8 | 5.1/2.0 | 19.6 | 5.0/3.2 |
| HSG $\rightarrow$ TSG $\rightarrow$ LSG | 16.9 | 5.1/2.1 | 19.5 | 4.9/3.2 |
| LSG $\rightarrow$ HSG $\rightarrow$ TSG | 17.2 | 6.3/2.0 | 19.8 | 4.8/2.7 |
| LSG $\rightarrow$ TSG $\rightarrow$ HSG | 17.1 | 5.4/2.0 | **19.0** | 5.0/2.7 |
| TSG $\rightarrow$ HSG $\rightarrow$ LSG | 16.9 | 5.5/1.8 | 19.3 | 5.0/3.2 |
| TSG $\rightarrow$ LSG $\rightarrow$ HSG | **16.7** | 4.9/2.1 | **19.0** | 4.9/2.9 |

# Table 14: Ablation experiments of SignGraph on PHOENIX14T dev set.

### (a) **Backbone**. Comparison of different backbones.

| BackBone | WER | Del | Ins |
|---|---|---|---|
| SwinT (52) | 45.4 | 16.3 | 1.3 |
| PyVIG (19) | 35.4 | 12.1 | 1.3 |
| SA (53) | 39.2 | 15.3 | 0.9 |
| Ours | **16.7** | 4.9 | 2.1 |

### (b) **Distance function**. Comparison of distance functions.

| Distance | WER | Del | Ins |
|---|---|---|---|
| Cosine | 17.4 | 5.5 | 2.0 |
| Chebyshev | 17.3 | 5.1 | 3.2 |
| Euclidean | **16.7** | 4.9 | 2.1 |

### (c) **Graph Convolution** Effect of different GCN layers.

| GraphConv | WER | Del | Ins |
|---|---|---|---|
| GATv2Conv (54) | 17.0 | 5.1 | 2.0 |
| SAGEConv (55) | 17.7 | 5.9 | 1.9 |
| GCNConv (49) | 17.7 | 5.5 | 2.0 |
| EdgeConv (56) | **16.7** | 4.9 | 2.1 |

### (d) **Patch size**. Comparison of different patch sizes with one $LSG$, $TSG$ and $BSG$ modules.

| PatchSize | WER | Del | Ins |
|---|---|---|---|
| 8 | 17.4 | 5.4 | 3.0 |
| 16 | 17.1 | 5.3 | 2.0 |
| 32 | 17.5 | 5.7 | 2.4 |

### (e) **Multiscale SignGraph**. Effect of the number of stages in multiscale SignGraph.

| Stages | WER | Del | Del |
|---|---|---|---|
| 16→32 | **16.7** | 4.9 | 2.1 |
| 8→16→32 | 17.1 | 5.2 | 2.0 |
| 4→8→16→32 | 17.8 | 5.4 | 3.1 |

### (f) **Drop edge**. Adding DropEdge (57) does not improve performance.

| DropRate | WER | Del | Ins |
|---|---|---|---|
| 0 | **16.7** | 4.9 | 2.1 |
| 15% | 17.7 | 6.1 | 1.4 |
| 30% | 17.5 | 6.0 | 2.1 |

node pairs) in the $TSG$ module, and fixed graphs (*i.e.*, connecting the same regions between feature maps) in the $HSG$ module. To verify the effectiveness of the above graph types, we evaluate the CSLR performance by changing the graph types in $LSG$, $TSG$ and $HSG$ modules. For simplicity, we only use one $LSG$ module, one $TSG$ module and one $HSG$ module in the experiment. As shown in Table 13a, when using a spare graph in the $LSG$ module, a dense graph in the $TSG$ module, or a dynamic dense graph (*i.e.*, building in the similar way with the $LSG$ module) in the $HSG$ module, there is a noticeable performance drop. That is to say, it is effective to adopt dense, sparse and fixed graphs in the $LSG$, $TSG$ and $HSG$ modules, and the combination of these mixed graphs can make our model achieve the best recognition performance.

**Effect of Ordering of Sign Graph Modules.** We also evaluate the CSLR performance by changing the ordering of three graph modules. As shown in Table 13b, different orderings of the graph modules lead to slight differences in the CSLR performance, *i.e.*, the ordering has a little effect on the overall effectiveness of feature extraction and integration. Nevertheless, the ordering $TSG \rightarrow LSG \rightarrow HSG$ can help the model to achieve the best CSLR performance. Thus our model MixSignGraph adopts the ordering $TSG \rightarrow LSG \rightarrow HSG$ by default.

**Effects of Different Backbones.** To verify the effectiveness of the MixSignGraph model, we replace our backbone with other patch-based networks (*e.g.*, CvT (58), Swin Transformer (52)), or replace our graph modules with Self-Attention (SA) layers (53). As shown in Table 14a, simply adopting the patch-based SOTA backbones or substituting our graph modules with SA layers does not lead to appealing performances on the CSLR task. In contrast, our proposed MixSignGraph backbone achieves a satisfactory performance by effectively capturing both cross-region and one-region features.

**Effect of Distance Function.** In the MixSignGraph, we use the KNN algorithm to find the nearest neighbors for a node/patch based on the distance between two nodes. To select a suitable distance

function to measure the distance between nodes, we evaluate the CSLR performance with the following commonly used distance functions, *i.e.*, Cosine distance, Chebyshev distance and Euclidean distance. As shown in Table 14b, there are subtle performance differences, when using different distance functions. In MixSignGraph, we adopt Euclidean distance for better performance.

**Effect of GCN Layer.** To verify the effectiveness of the graph convolution layer, we test the CSLR performance by adopting the following representative variants of graph convolution, *i.e.*, GATv2Conv (54), SAGEConv (55), GCNConv (49), and EdgeConv (56). As shown in Table 14c, even using different GCN layers, our model still achieves good performance. It indicates that our MixSignGraph model has good flexibility in GCN layer selection. Nevertheless, among the above GCN layers, EdgeConv can help our model to achieve the best CSLR performance, *i.e.*, 16.7% WER. Therefore, we adopt the EdgeConv layer in MixSignGraph for better performance.

**Effect of Patch Sizes.** Patches with different sizes have different receptive fields, thus can capture different-range features and further affect the model performance. To observe the effect of patch sizes, we evaluate the CSLR performance by only adopting one $LSG$ module, one $TSG$ module and one $HSG$ module, while changing the patch size. As shown in Table 14d, when using a patch size that is too small (*i.e.*, 8×8) or too large (*i.e.*, 32×32), the model MixSignGraph shows a worse performance. We found that a suitable patch size strikes the best balance, leading to good performance.

**Effect of Multiscale Sign Graphs.** To leverage the scale-invariant property of images (19), PyVIG adopts a pyramid architecture that gradually increases patch size from 4 to 32 by shrinking the spatial size of feature maps to enhance feature extraction capability. Similarly, to extract better sign features and achieve higher CSLR performance, we gradually add $LSG$, $TSG$ and $HSG$ modules after the early stage of the patchify stem. According to Table 14e, adding more $LSG$, $TSG$ and $HSG$ modules does not always bring performance gains. Therefore, MixSignGraph adopts two $LSG$, $TSG$ and $HSG$ modules, *i.e.*, increasing patch size from 16 to 32, aiming to achieve the best performance.

**Effect of DropEdge.** DropEdge (57) can alleviate the over-smoothing and overfitting problems in dense graphs by randomly removing a certain number of edges from the input graph at each training epoch. Considering that we build dense graphs for the $LSG$ and $HSG$ modules, we introduce DropEdge into these two modules and change the drop rates of DropEdge to evaluate the CSLR performance. As shown in Table 14f, adding DropEdge does not improve the performance of our MixSignGraph. It indicates that our model can achieve excellent performance without relying on external modules (*e.g.*, DropEdge) and it can handle over-smoothing and overfitting problems well.

