# OpenReview forum: "MixSignGraph: A Sign Sequence is Worth Mixed Graphs of Nodes"
_NeurIPS.cc/2025/Conference — NeurIPS 2025 poster_

### Official Review · Reviewer_zjcq · 2025-06-11

**Clarity:** 3
**Significance:** 3
**Originality:** 3
**Rating:** 5
**Confidence:** 4

**Summary:**

This paper introduces MixSignGraph, which represents a sign sequence as mixed graphs of nodes. Compared to the previous work, termed as MultiSignGraph, it makes two key changes. The first is a Hierarchical Sign Graph module, and the second is Text-based CTC Pre-training strategy.

**Questions:**

See above

**Ethical Concerns:**

["NO or VERY MINOR ethics concerns only"]

**Final Justification:**

Considering the rebuttal, i think this paper is novel with sufficient experiemental results. I will maintain my positive rating and lean towards accepting this paper.

**Limitations:**

Yes

**Quality:**

3

**Strengths And Weaknesses:**

Pros:
1. The paper is written in a clear and concise manner, making it easy to follow the proposed methodology, experimental setup, and results. The logical organization enhances readability and ensures that the contributions are effectively communicated to the reader.
2. The experimental evaluation demonstrates strong performance across multiple benchmarks, suggesting that the proposed method is effective and competitive with existing approaches. These results support the claim that the method offers practical benefits in real-world applications.
3. The proposed approach introduces several novel components that distinguish it from previously published methods. These innovations contribute meaningfully to the field and offer new directions for future research in multi-modal or hierarchical learning frameworks.

Cons:
1. While the paper employs different backbone architectures compared to baseline methods, it does not provide any discussion or comparison regarding computational costs—such as training time, inference speed, or model complexity. Including such an analysis would help readers better understand the trade-offs between performance gains and resource requirements.
2. In Table 1, the meanings of the labels "1" and "2" are not explained. This lack of clarification makes it difficult for readers to interpret the experimental variations being compared. A brief explanation or footnote in the table would significantly improve its clarity and usefulness.

Minor questions:
1. It would be helpful to clarify how the first Hierarchical Sign Graph module behaves when the input feature vector $u_l$  is absent. Does the module skip processing at that level, or is there a default mechanism in place? Understanding this edge case would improve reproducibility and robustness assessment.
2. In Table 2, the abbreviation "TCP" appears without prior definition. Please clarify what "TCP" stands for to avoid confusion and ensure accurate interpretation of the experimental settings.
3. If certain datasets include gloss annotations, would the authors still employ the proposed Text-based CTC Pre-training strategy? It would be interesting to know whether the presence of such annotations influences the choice of pre-training method, and if so, how.

---

> ### Author Rebuttal · Authors · 2025-07-31
>
> We appreciate your insightful comments, which have helped strengthen our work. Below, we provide individual responses to your questions.
>
> ## Q1: Computational costs
>
> We actually did analyze the computational complexity of our proposed MixSigngraph in Appendix: Table 10 and Section A.5 (on page 15). Besides, we also provide training time and inference speed in the following table, where the metric was measured with a 3090 GPU on 280 frames.
>
> Compared with SIGN2GPT, TwoStream, our model achieves faster inference speed (translates about 2 SL video sequence per second), while achieving the promising performance on multiple SL tasks, as shown in the table below and Tabel 10 on page 15.
>
>
> |model|backbone|Inference Speed|Traning time|
> |-|-|-|-|
> SIGN2GPT| Dino-V2 ViT|1 seq/s|-
> TwoStream |Dual S3D|1.4 seq/s|-
> SignGraph |GCN|2.1 seq/s| 44 min/epoch
> MixSigngraph |GCN|2.0 seq/s| 45min/epoch
>
> ## Q2: Unclear footnote  in Table 1,2
>
> Thank you for pointing this out! We apologize for the unclear footnote in our Table. $LSG_1$, $TSG_1$ and $HSG_1$ means the first $LSG$, $TSG$, $HSG$ module, and $LSG_2$, $TSG_2$ and $HSG_2$ means the second $LSG$, $TSG$, $HSG$ module.  We will thoroughly review the manuscript and correct it, ensuring the ideas are conveyed more clearly and professionally in the final version. We appreciate your patience and feedback.
>
>
> ## Q3: How does the first Hierarchical Sign Graph module behave when the input feature vector is absent?
>
> As shown in Figure 2, Patchify Steam consists of convolution layers. Before downsampling to get $u_0^l$, we first generate  $u_0^h$, which is one of the initial input features $u_i^h$(i=0) of the first HSG.  As for $u_0^l$, after being processed by the LSG and TSG, it will be fed into the HSG as another inital input of first HSG.
> We will add a more detailed description, ensuring the ideas are conveyed more clearly.
>
> ## Q4: "TCP" appears without prior definition.
>
> We apologize for the misspelling. 'TCP' should be written as the "TCTC" module. We will correct this error in the final version.
>
>
> ## Q5: Use of Text-based CTC Pre-training with Datasets Containing Gloss Annotations
>
> When gloss annotations are available (e.g., dataset:CSL-daily and Phoenix14T), we evaluate our model on two different tasks: gloss-based SLT and gloss-free SLT.
>
> - **In the gloss-based SLT task**, TCTC is not employed. Instead, the ground truth (GT) gloss sequence is used to pretrain our backbone model.
>
> - **In the gloss-free SLT task**, gloss annotations are **not** used for training. Instead, **TCTC** is employed to generate pseudo glosses for pretraining the backbone model.
>
>
> In Tables 5 and 6, we present both the gloss-based SLT performance (Sign2Text) and the gloss-free SLT performance (Gloss-free SLT) for CSL-daily and Phoenix14T. Generally, the gloss-based SLT models perform better than the gloss-free SLT models on both datasets.
>
>
> In real-world scenarios, if gloss annotations are available, it is preferable to choose the ground truth gloss to pretrain the model in order to achieve better performance. We hope this clarifies how the presence of gloss annotations influences our choice of pre-training method.
>
> We appreciate your questions and comments. Please let us know for any further questions.

---

> > ### Comment · Reviewer_zjcq · 2025-08-05
> > **Response**
> >
> > After reading the rebuttal, i will maintain my positive rating and lean towards accepting this paper.

---

> ### Author Response · Authors · 2025-08-05
>
> Thank you for taking the time to review our manuscript and responses, as well as for your insightful comments that will greatly help us improve our work.
>
> We greatly appreciate your recognition of our work and your decision to maintain  the positive score.

---

### Official Review · Reviewer_yMHm · 2025-06-24

**Clarity:** 2
**Significance:** 2
**Originality:** 2
**Rating:** 4
**Confidence:** 3

**Summary:**

The paper proposes MixSignGraph, a novel GCN-based framework for continuous sign language recognition (CSLR) and translation (SLT). The introduced novelty is mainly two-fold: i) MixSignGraph introduces a new Hierarchical Sign Graph (HSG) module that combines features at different levels of granularity, allowing the model to focus on both global and local shapes at the same time; and ii) it introduces Text-Based CTC Pre-training (TCTC) for gloss-free SLT tasks, where pseudo gloss sequences are built from translations by applying traditional NLP techniques like stemming and lemmatization.

**Questions:**

Please address the weaknesses I mentioned above, in particular those related with the scaling of the method to long sequences and the evaluation metrics. If these are addressed appropriately and a writing review is performed to clear the typos, I am willing to raise my score.

**Ethical Concerns:**

["NO or VERY MINOR ethics concerns only"]

**Final Justification:**

Having read the rebuttal and the other reviews, I still lean towards acceptance. However, I wouldn't disagree if the decision went the other way, as this would enable you to make all the necessary changes to your manuscript and submit an improved version to another venue. Thus, I will keep my score (4).

**Limitations:**

Yes.

**Paper Formatting Concerns:**

Only typos.

**Quality:**

3

**Strengths And Weaknesses:**

Strengths:
- The HSG design is intuitive and effectively addresses a clear limitation of prior work.
- The TCTC pre-training is an interesting practical workaround for gloss-free settings.
- The experimental results consistently show improvements across CSLR and SLT tasks, with a performance competitive or superior to the state of the art.
- The proposed approach is modality efficient, as it achieves high performance only from RGB data.

Weaknesses:
- Although the HSG design is intuitive, it lacks formal justification or theoretical analysis of why it improves representation.
- The use of GCNs and multiple graph modules (LSG, TSG, HSG) might raise scalability issues on long video sequences. This issue is not analyzed nor is it mentioned in the limitations section.
- Only lexical overlap metrics (ROUGE and BLEU) are used for evaluation. This is especially problematic in SLT, where there’s often no one-to-one correspondence between signs and words, and translations tend to be semantically but not lexically equivalent.
- The writing has many typos (e.g. "Limitations and *Borader* Impacts"), which is not acceptable. These sometimes make the idea hard to follow. The authors should definitely fix these.

---

> ### Author Rebuttal · Authors · 2025-07-31
>
> We appreciate your insightful comments. Below, we provide individual responses to your questions.
>
> ## Q1: Lack formal justification or theoretical analysis
>
> Thank you for the comment. Here we try to provide a brief formal analysis for our HSG in MixSignGraph and discuss why HSG can boost discriminative local features.
>
> - For each frame $f_i\in \mathbb{R}^{[H\times W \times 3]}$ with height $H$ and width $W$, we get a number of patches (or nodes) $v _ i= \\{v _ {ij} \\} _ {j=1}^{N}$ with its feature $\mu _ {ij}$.
>
> - Local Sign Graph (LSG) can be simplified as: $\mu _ i =  \mathcal{GCN}_L(\mu _ i, e _ i^l) $ where $e _ i^l = \{ e(v _ {ij}, v _ {ik}) \mid j \in [0, N), v _ {ik} \in \mathcal{N} _ {\mathsf{K} _ {l}}(v _ {ij}) \}$ and $\mathcal{N} _ {\mathsf{K} _ {l}}(v _ {ij}) =\{v _ {ik} | v _ {ik} \in  TopK (\{\mathcal{DIS}(v _ {ij}, v _ {ik})\} _ {k=1}^{N}) \}$. LSG extracts intra-frame cross-region features by aggregating neighbor $\mathcal{N} _ {\mathsf{K} _ {l}}(v _ {ij}) $ features within frames based on node similarity.
>
> - Temporal Sign Graph (TSG) can be simplified as: $\\{\mu _ i\\}^{\theta}_{i=1}= \mathcal{GCN}_t\left ( \\{\mu _ i\\}^{\theta} _ {i=1},  \\{e _ i^t\\}^{\theta-1} _ {i=1} \right) $ where $e^t _ {i}=\\{e(v _ {ij},v _ {(i+1)k}) |  \mathcal{DIS}(v _ {ij}, v _ {(i+1)k} ) \in TopK(M)\\}$, $M$ denotes similarity matrix. TSG extracts inter-frame cross-region features by aggregating neighbor features between frames based on node similarity.
>
> - Suppuse orginal node $\mu_{ij}$ follows a Gaussian distribution:
> $\mathcal{G}(\chi _ {ij}^{(n)}, \sigma _ {ij}^{(n)}), n \in [1, N]$. Then $\mu _ {ij}^{new} \to \mathcal{G}(\bar{\chi}, \bar{\sigma})$ , where $\bar{\chi}$, $\bar{\sigma}$ represents the weighted average of $\mathcal{N}(\chi _ {ij}^{(n)})$, $\mathcal{N}(\sigma _ {ij}^{(n)})$. $\mathcal{N}$ denotes its neighbors. The difference between two neighbor nodes cloud be: $\mathbb{E}[||\mu _ {ij}^{new} -\mu _ {ik}^{new} ||^2] \to 0$
>
> Although, the LSG and TSG modules facilitate the extraction of cross-region features. However, the nodes within neighboring regions may become overly similar, especially when there are too many edges or the GCN layers are too deep. This leads to the over-smoothing phenomenon [1,2], which weakens the distinctiveness and location sensitivity of a local region of $v _ {i,j}$.
>
>
>
> - While HSG, in constract: $\mu _ {i}^{h'}, \mu _ {i}^{l'}= \mathcal{GCN} _ H(\{\mu _ {i}^{h}, \mu _ i^l\}, e _ b^i)$ where $e _ i^b = \\{ e(v _ {ij}^h, v _ {ik}^l)|j \in  [0,N^h),
> k = ( \begin{matrix}\left \lfloor \frac{\left \lfloor \frac{j}{W^h} \right \rfloor}{s} \right \rfloor \end{matrix}  \times  W^l + \left \lfloor \frac{j\%W^h}{s} \right \rfloor )  \\}$, and $\mu _ i^h$ refers to the feature map before cross-region extraction, which well preserves the integrity of local features and structure.
> HSG aggregates features in the same regions of $\mu _ i^h$ and $\mu _ i^l$. Therefore, the local information of nodes can be effectively enhanced, and then the over-smoothing phenomenon is alleviated.
> That is to say, the hierarchical interaction between same-region features in different granularities ensures that both distinctiveness and location of local features are maintained, improving the model’s overall performance.
>
> We will provide a more detailed analysis and add it in our paper.
>
> [1] Wu, Xinyi, et al. "Demystifying oversmoothing in attention-based graph neural networks." Advances in Neural Information Processing Systems 36 (2023): 35084-35106.
>
> [2]Wu, Xinyi, et al. "A Non-Asymptotic Analysis of Oversmoothing in Graph Neural Networks." The Eleventh International Conference on Learning Representations.
>
>
> ## Q2: Scalability Issues with Graph Modules
>
> Thank you for raising the concern about the potential scalability issues with LSG, TSG, HSG on long video sequences. We appreciate your insight, and we would like to clarify the following points:
>
> - LSG and HSG are built based on  single frames, which are not limited by the video length. These modules focus on aggregating features at the frame level; thus, they do not introduce scalability issues for long video sequences.
>
> - TSG is designed to connect **adjacent frames** in the sequence. While MixSignGraph currently builds the TSG on the entire video, this may pose challenges for modeling long videos. To address this, we have conducted an ablation study to explore the effect of connecting different numbers of frames in TSG (i.e., connecting every N frames temporally in TSG instead of collecting all frames).  The model's performance remains relatively stable regardless of how many frames are connected in TSG.
>
> Therefore, for long videos, instead of constructing a graph over the entire video, we can group the video into smaller temporal segments, and this grouping does not affect the model’s performance, confirming that scalability is not an issue for long video sequences.
>
> | Backbone | WER  | Del/Ins |
> |----------|------|---------|
> | Whole    | 16.73 | 5.1/2.0 |
> | N=10       | 16.72 | 5.0/1.5 |
> | N=50       | 16.75 | 5.3/2.2 |
> | N=100      | 16.72 | 5.2/1.3 |
>
> We will add this analysis in the final version of our paper.
>
>
>
>  ## Q3: Evaluation metrics
>
> Thank you for the constructive comment regarding evaluation metrics.
>
> - **Adoption of ROUGE and BLEU**: We mainly followed previous SLT work and adopted the widely used metrics ROUGE and BLEU, to maintain a fair comparison with state-of-the-art models.
>
> - **Semantic Evaluation**: We acknowledge that the above metrics may not fully capture the semantic meaning of translations.  As you pointed out, ROUGE and BLEU are based on lexical overlap, which may fail to evaluate the **semantic quality** of translations. For example, given a **ground truth** (GT: "I like dog.") and two possible candidates (c1: "I love dog." and c2: "I hate dog."), Both predictions may have the same **ROUGE** and **BLEU** scores (Rouge(c1, GT) $\approx $ 66.6), Rouge(c2, GT) $\approx $ 66.6), even though "I love dog" is clearly more semantically accurate.
>
> To address this issue, we follow the approach proposed in [3] and additionally adopt **BLEURT**, **CIDEr**, and **LLM** scores to better capture the semantic quality of our model’s outputs.
>
> - **BLEURT[1]**: A trained metric that uses a regression model, which is trained on human ratings data. It captures non-trivial **semantic similarities** between sentences.
>
> - **CIDEr[2]**: A captioning metric that computes the **consensus** of the prediction compared to the ground truth by calculating the weighted cosine similarity of TF-IDF scores for various n-grams.
> - **LLM[3]**: An LLM-based evaluation metric proposed by [3]. We use the publicly available **GPT-4o-mini API** from OpenAI, prompting the model to generate a score from **0 to 5** for each translation pair to measure the matching degree between predicted sequence and ground-truth sequence, where 5 indicates the best match and 0 indicates  the worst.
>
> The results for these additional evaluation metrics are shown below:
>
> | Dataset    | BLEURT $\uparrow$ | CIDEr  $\uparrow$ | LLM  $\uparrow$ |
> |------------|--------|-------|------|
> | Phoenix14T(Ours) | 59.68  | 240.23| 3.1  |
> | CSL-daily(Ours)  | 57.47  | 233.23| 3.0  |
> |how2sign(Ours)|50.1 | 120.4| 1.9|
> |how2sign(Lost[3])| 45.3|  100.8 | 1.59|
>
> We will provide a more comprehensive assessment of our model's semantic performance. More detailed results will be included in the final version of the paper.
>
> [1] Sellam, Thibault, Dipanjan Das, and Ankur Parikh. "BLEURT: Learning Robust Metrics for Text Generation." Proceedings of the 58th Annual Meeting of the Association for Computational Linguistics. 2020.
>
> [2] Vedantam, Ramakrishna, C. Lawrence Zitnick, and Devi Parikh. "Cider: Consensus-based image description evaluation." Proceedings of the IEEE conference on computer vision and pattern recognition. 2015.
>
> [3] Jang, Youngjoon, et al. "Lost in translation, found in context: Sign language translation with contextual cues." Proceedings of the Computer Vision and Pattern Recognition Conference. 2025.
>
>
> ## Q4: Typos
>
> Thank you for pointing this out!  We will thoroughly review the manuscript and correct all typos (e.g., "Borader" to "Broader"),  ensuring the ideas are conveyed more clearly and professionally in the final version. We appreciate your patience and feedback.
>
> We appreciate your questions and comments. Please let us know for any further questions.

---

> ### Comment · Reviewer_yMHm · 2025-08-04
> **Answer to Authors' rebuttal**
>
> Dear Authors,
>
> Thank you for your hard work and detailed answers to my questions. I believe including this additional content in your manuscript will improve its value. Addressing your answers one by one:
>
> Q1: I think this content should be expanded and included in the manuscript for review. For this reason, your manuscript might benefit from another round of review in a future venue.
>
> Q2: Your answer solved my concerns.
>
> Q3: Although I admire your effort to have these results ready in time for the rebuttal, the comparison with other methods that you provide in your answer to Q3 is still limited.
>
> Q4: Your answer solved my concerns.
>
> Having read your rebuttal and the other reviews, I still lean towards acceptance. However, I wouldn't disagree if the decision went the other way, as this would enable you to make all the necessary changes to your manuscript and submit an improved version to another venue. Thus, I will keep my score (4).

---

> ### Author Response · Authors · 2025-08-05
>
> Thank you for taking the time to review our responses, as well as for your insightful comments that will greatly help us improve our work. Here, we provide a more comprehensive comparison.
>
> ## More comparisons with other methods
>
> ### Evaluation metrics
> - As previously mentioned, most existing SLT works primarily adopt ROUGE and BLEU scores as evaluation metrics. BLEURT, CIDEr, and LLM-based scores are not widely reported in most existing papers.
> On CSL-Daily and Phoenix14T, previous methods mainly report ROUGE and BLEU, which limits the scope of comparison.  Therefore, we conduct the following comprehensive comparisons on How2Sign and OpenASL.
>
>
> ### Training Setup
> - For How2Sign and OpenASL, we use a pretrained I3D for feature extraction (see Section 4: Training Settings) to speed up training in the main paper. Importantly, I3D is not trained during the process, and this may lead to lower performance. Now, we further provide the full training results (i.e., Our(Full)) to show the better performances achieved by MixSignGraph.
>
>
> ### Compared Methods
>
> - We make comparsions across multiple metrics, including semantic-based scores (BLEURT [1], CIDEr [2], LLM [3]) and lexical overlap scores (ROUGE, BLEU1, BLEU4).  It is worth noting that methods marked with asterisks (e.g., SSVP-SLT*, SHuBERT*, Scaling*) leverage large-scale external SL datasets (e.g., YT-ASL, YT-Full) for pretraining, contributing to their strong performance. In particular, Scaling* [8] also utilizes the Multilingual Machine Translation dataset, enabling it to achieve SOTA results. However, it may be unfair to directly compare our model with these models, as our model does not rely on any extra SL datasets for pretraining.
>
> ### Performance Analysis
>
> - As shown in the Table 1, on the How2Sign , MixSignGraph with full training (Our(Full)) usually outperforms the non-pretrained baselines(i.e., SSVP-SLT, Lost, GloFE-VN, Scaling). Besides, without using external SL datasets for pretraining, our method (Our(Full)) outperforms SSVP-SLT* and SHuBERT*, and remains competitive with Scaling* in terms of BLEURT score.
> When moving to Table 2, on the OpenASL, our model  (Our(Full)) surpasses GLoFE-VN by a significant margin (45.09 vs. 36.35 on BLEURT), further demonstrating the effectiveness the proposed MixSignGraph.
>
> **Table1: Comparsion  on How2Sign test set.**
>
> | How2Sign    | BLEURT $\uparrow$ | CIDEr  $\uparrow$ | LLM  $\uparrow$ | ROUGE|BLEU1|BLEU4|
> |-|-|-|-|-|-|-|
> |SSVP-SLT [7]|39.3|-|-|25.7|30.2|7.0|
> |Lost [3]| 45.3|  **100.8** | **1.59**|32.5|**45.3**|12.7  |
> |GloFE-VN [4]|31.65|-|-|12.61|14.94|2.24|
> |Scaling [8]|34|-|-| ||-|
> |SSVP-SLT* [7]|49.6|-|- |**38.4**|**43.2**|15.5|
> |SHuBERT* [6]|49.9|||||**16.2**|
> |Scaling* [8]|**51.74**|-|-| ||**21.06**|
> |Our (I3D)|40.0 | 100.1| 1.4|28.01|34.74|10.41|
> |Our (Full)|**50.1** | **120.4** | **1.9** |**39.46**|39.56|14.6|
>
>
> **Table 2: Comparsion on OpenASL test set.**
>
> | OpenASL    | BLEURT $\uparrow$ | CIDEr  $\uparrow$ | LLM  $\uparrow$ |ROUGE|BLEU1|BLEU4|
> |-|-|-|-|-|-|-|
> GloFE-VN [4]|36.35|-|- |21.75|21.56|7.06
> Our (I3D) | 36.11| 102.1|1.30|25.7|26.65|8.69
> Our (Full)| **45.09**| **130.3**|**1.78**|**42.49**|**39.61**|**15.61**
>
> **Note: Methods marked with an asterisk * use large-scale SL datasets for pretraining.**
>
> ## Case Analysis
> -  To intuitively demonstrate that the new metrics better capture semantic similarity, we present a sample case comparing semantic-based and lexical overlap scores. Given the ground truth (GT:"I like dogs.") and two predictions (P1: "I love dogs." , P2:"I hate dogs."), P1 is clearly more semantically accurate.
>
> - As can be seen, although P1 and P2 achieve the same BLEU1 and ROUGE scores, P1 achieves higher BLEURT and LLM scores. This highlights the limitation of lexical metrics in capturing semantic differences, while BLEURT and LLM scores better reflect semantic quality.
>
> |Example|BLEURT $\uparrow$ |LLM $\uparrow$ |CIDEr |BLEU1|ROUGE|
> |-|-|-|-|-|-|
> |P1|71.96|4|-|66.67|66.67
> |P2|-1.5|1|-|66.67|66.67
>
> **Note: CIDEr is a corpus-level metric and cannot be computed on individual sentences.**
>
> [1] Sellam, et al. BLEURT: Learning Robust Metrics for Text Generation. In ACL 2020.
>
> [2] Vedantam, et al. Cider: Consensus-based image description evaluation. In CVPR 2015.
>
> [3] Jang, et al. Lost in translation, found in context: Sign language translation with contextual cues. In CPVR 2025.
>
> [4] Lin, et al. Gloss-Free End-to-End Sign Language Translation. In ACL 2023.
>
> [5] Gueuwou, et al. Signmusketeers: An efficient multi-stream approach for sign language translation at scale. In ACL 2024.
>
> [6] Gueuwou, et al. SHuBERT: Self-Supervised Sign Language Representation Learning via Multi-Stream Cluster Prediction. In CoRR 2024.
>
> [7] Rust, et al. Towards Privacy-Aware Sign Language Translation at Scale. In ACL 2024.
>
> [8] Zhang, et al. Scaling sign language translation.  In NeurIPS 2024.
>
> We appreciate your comments. Please let us know for any further questions.

---

### Official Review · Reviewer_rodL · 2025-07-02

**Clarity:** 3
**Significance:** 3
**Originality:** 2
**Rating:** 4
**Confidence:** 4

**Summary:**

The paper introduces a new method named MixSignGraph for sign language recognition from videos. The method is an extension to SignGraph [4] that introduces local sign graph and temporal sign graph for sign language recognition. The proposed method adds an additional hierarchical sign graph to capture correlations between visual features. Furthermore, the paper introduces Text-based CTC Pre-training to further improve the performance.

**Questions:**

Please see weaknesses above regarding questions.

**Ethical Concerns:**

["NO or VERY MINOR ethics concerns only"]

**Final Justification:**

The authors have addressed most of my concerns with the rebuttal.

**Limitations:**

Yes

**Quality:**

3

**Strengths And Weaknesses:**

Positives:

-	The paper shows that by adding the hierarchical sign graph, the proposed method can improve the sign language recognition performance.

-	The paper has conducted extensive experiments on several datasets and alation studies to verify the proposed method.

Negatives:

-	Since the method is an extension of the SignGraph [4] method, the novelty and technical contribution of the method is limited. The main difference is by introducing the hierarchical sign graph into the method.

-	Although the Text-based CTC Pre-training shows improvements, however, the model may overfit to these datasets by using the texts in the datasets for pre-training. As a result, the generalization ability of the method may be reduced.

-	In tables 3, 4, 5, 6, 7, why not comparing to the original SignGraph [4] method?

-	Models such as the Swin transformer also utilize hierarchical representations. Will a Swin transformer perform as good as the hierarchical sign graph in the proposed method?

---

> ### Author Rebuttal · Authors · 2025-07-31
>
> We appreciate your thoughtful comments. Below we provide answers to the comments you raised.
>
> ## Q1: The novelty and contribution are limited. The main difference is in introducing the HSG.
>
> We would like to clarify that our work introduces two substantial and original contributions that go beyond a simple architectural refinement.
>
> (1) **Identifying and addressing a core limitation of SignGraph**: SignGraph builds LSG and TSG modules to model the intra-frame and inter-frame cross-regions features. While cross-region interactions have been shown to improve CSLR performance, we find that cross-region interactions may also weaken the feature representation and location sensitivity of a single/local region.
>
> - (a) The distinctiveness of a local region’s representation is disrupted and weakened by features from other connected regions.
>
> - (b) The spatial structure of regions (i.e., the location of a region, such as the hand area, within a frame) is also weakened by the interactions with other connected regions.
>
> To alleviate the above issue, we propose the Hierarchical Sign Graph module, which aggregates the same-region features from different-granularity feature maps of a frame. By highlighting the features in a single region, which is located at a specific position within a frame, HSG is possible to enhance the distinctiveness and location sensitivity of a local region itself, even after cross-region feature extraction. This is not merely an incremental change, but a theoretical and technical rethinking of how region interactions should be modeled in sign sequences.
>
>
> (2) **Identifying and addressing a core limitation of Gloss-Free SLT (GFSLT)**:  Additionally, our approach extends beyond the task scope of SignGraph. While SignGraph focuses solely on CSLR tasks, our MixSignGraph can address both CSLR and SLT tasks(including Sign2Text, Sign2Gloss2Text, and GFSLT).
>
> More importantly, in SLT tasks, we find that the current **bottleneck in Gloss-Free SLT** lies in the **continuous feature distribution** of SL videos, which does not align with the **discrete feature distribution** required by translation models, where the inputs are typically discrete tokens (see Figure 4 and Section 3.1). In contrast, gloss-based SLT benefits from gloss annotations with CTC constraints to effectively discretize SL features. Building on this insight, we propose a TCTC pretraining method, which leverages pseudo gloss sequences and CTC to guide the discretization of SL video features.
>
>
> ## Q2: TCTC may overfit to these datasets, the generalization ability may be reduced?
>
> - It is worth noting that text annotations are essential for training an SLT model, whether gloss-based or gloss-free, and regardless of whether the TCTC module is adopted. Our method does not introduce additional text annotations for training (i.e., it uses the same video-text training samples as other gloss-based and gloss-free SLT models). Instead, we focus on improving how text is used during the training stage by generating pseudo glosses to pretrain the model.
>
> - We also highlight that the principle behind TCTC (as detailed in Section 3.1) is that pseudo glosses, generated from the text, are used to discretize continuous SL features, thereby providing suitable input for translation models, whose inputs are typically discrete tokens (see Figure 4 and Section 3.1). Importantly, **TCTC is not designed to fit the text in the datasets**. Instead, it serves as a form of pseudo-supervision for the visual encoder, guiding  it to tokenize sign language videos effectively. The goal is not to directly fit the model to the training text, but to provide discrete and meaningful representations for SLT.
>
> - The comparison results of dev/test set on four datasets (Single domain: CSL-daily, Phoenix14T. Open domain: How2sign, Openasl) demonstrate our TCTC's generalization. TCTC maintains good generalization ability across different domains, further proving its effectiveness.
>
>  ## Q3: Comparing to the original SignGraph method?
>
> In fact, we have compared our method with SignGraph. Specifically, the SignGraph paper proposed two models: SignGraph and the enhanced version MultiSignGraph. In the main paper, we have presented comparisons with MultiSignGraph, which achieves better performance than SignGraph, as shown in Tables 3, 4, 5, 6, and 7. To provide a more comprehensive comparison, we will add the comparison with SignGraph in the final version.
>
>
>
>
> ## Q4: Comparing with Swin transformer
>
> We have already compared Swin Transformer (SwinT) with our method in Appendix: Section A.7 and Table 14, which is in page 17 of our paper. We also list results here for clarity:
>
> |Backbone|WER|Del/Ins|
> |-|-|-|
> |SwinT |45.4| 16.3/1.3
> |PyVig|35.4|12.1/1.3
> |SelfAttention|39.2|15.3/0.9
> |ours|16.7|5.1/2.0
>
> As shown in the table, the Swin Transformer and PyVig, which also utilize hierarchical structures, perform worse than our method. This highlights that MixSignGraph with HSG can effectively enhance local region representations following the extraction of cross-region features, addressing the limitations of SignGraph and further improving SLT performance.
>
> We appreciate your questions and comments. Please let us know for any further questions.

---

> > ### Comment · Reviewer_rodL · 2025-08-04
> > **After rebuttal**
> >
> > The authors have addressed most of my concerns. I am willing to raise my score for this paper.

---

> > > ### Author Response · Authors · 2025-08-04
> > >
> > > Thank you for taking the time to review our manuscript and responses, as well as for your insightful comments that will greatly help us improve our work.
> > >
> > > We greatly appreciate your recognition of our work and your decision to raise the score.

---

### Official Review · Reviewer_MqJV · 2025-07-02

**Clarity:** 2
**Significance:** 2
**Originality:** 2
**Rating:** 2
**Confidence:** 5

**Summary:**

This paper proposes MixSignGraph, a novel graph neural network architecture for sign language tasks, including Continuous Sign Language Recognition (CSLR) and Sign Language Translation (SLT). Building upon SignGraph, the authors introduce a Hierarchical Sign Graph (HSG) module to enhance local representations by aggregating same-region features across multi-scale feature maps. This design mitigates the degradation of local information typically caused by cross-region feature fusion. Furthermore, the authors present a simple yet effective Text-based CTC Pre-training (TCTC) strategy for gloss-free SLT, which generates pseudo gloss labels from raw text to guide model pre-training. Extensive experiments on five public datasets demonstrate the model’s superiority over existing state-of-the-art (SOTA) approaches, without relying on any additional modalities or annotations.

**Questions:**

Plz refer to the weaknesses.

**Ethical Concerns:**

["NO or VERY MINOR ethics concerns only"]

**Final Justification:**

While the paper introduces a promising framework, its core contributions remain incremental, and key claims—particularly regarding TCTC's effectiveness in gloss-free SLT—are not supported by sufficiently rigorous or fair comparisons. The evaluation emphasizes less meaningful metrics (e.g., ROUGE over BLEU-4), and recent relevant SOTA works are either overlooked or inadequately addressed. Additionally, if the focus is on improving visual representations, SLT may not be the most appropriate task for validation. For these reasons, I lean toward rejection.

**Limitations:**

Yes

**Quality:**

2

**Strengths And Weaknesses:**

Strengths

1. Effective Enhancement of Local Representations via HSG:
The HSG module strengthens local feature quality by establishing hierarchical connections across multi-resolution feature maps, complementing spatial and temporal cross-region modeling.

2. Innovative Gloss-Free Pre-training Strategy (TCTC):
The TCTC method enables pre-training using only textual labels, significantly improving gloss-free SLT performance and narrowing the gap with gloss-supervised models.


Weaknesses

1. Incremental Contribution over SignGraph.
While the proposed method introduces performance gains by adding modules such as the Hierarchical Sign Graph (HSG) on top of SignGraph, the technical novelty appears limited. The framework largely builds upon prior work, and the architectural modifications, though effective, may not be conceptually groundbreaking.

2. Insufficient Comparison with Recent State-of-the-Art.
The paper lacks comprehensive comparisons with several recent state-of-the-art models (e.g., [1, 2, 3]). Some of these models, including those that rely solely on pose or skeleton data, report superior or comparable results, yet are not adequately discussed or benchmarked against MixSignGraph.

[1] Li, Zecheng, et al. "Uni-Sign: Toward Unified Sign Language Understanding at Scale." arXiv preprint arXiv:2501.15187 (2025).

[2] Jiao, Peiqi, Yuecong Min, and Xilin Chen. "Visual Alignment Pre-training for Sign Language Translation." European Conference on Computer Vision. Cham: Springer Nature Switzerland, 2024.

[3] Chen, Zhigang, et al. "C 2 RL: Content and Context Representation Learning for Gloss-free Sign Language Translation and Retrieval." IEEE Transactions on Circuits and Systems for Video Technology (2025).

3. Semantic Inaccuracy in Pseudo Glosses.
The pseudo glosses used in TCTC are generated via basic linguistic preprocessing (e.g., lemmatization and tokenization), which may fail to capture the true semantics and structure of authentic gloss annotations. The paper should provide a more detailed analysis of how such semantic discrepancies affect performance, especially in gloss-free SLT tasks.

---

> ### Author Rebuttal · Authors · 2025-07-31
>
> Thank you for your insightful comments. Please find our point-by-point responses below.
>
> ## Q1: Incremental Contribution
>
> We would like to clarify that our work introduces two substantial and original contributions that go beyond a simple architectural refinement.
>
>
> (1) **Identifying and addressing a core limitation of SignGraph**: SignGraph builds LSG and TSG modules to model the intra-frame and inter-frame cross-regions features. While cross-region interactions have been shown to improve CSLR performance, we find that cross-region interactions may also weaken the feature representation and location sensitivity of a single/local region.
>
> - (a) The distinctiveness of a local region’s representation is disrupted and weakened by features from other connected regions.
>
> - (b) The spatial structure of regions (i.e., the location of a region, such as the hand area, within a frame) is also weakened by the interactions with other connected regions.
>
> To alleviate the above issue, we propose the Hierarchical Sign Graph module, which aggregates the same-region features from different-granularity feature maps of a frame. By highlighting the features in a single region, which is located at a specific position within a frame, HSG is possible to enhance the distinctiveness and location sensitivity of a local region itself, even after cross-region feature extraction. This is not merely an incremental change, but a theoretical and technical rethinking of how region interactions should be modeled in sign sequences.
>
>
>
> (2) **Identifying and addressing a core limitation of Gloss-Free SLT (GFSLT)**:  Additionally, our approach extends beyond the task scope of SignGraph. While SignGraph focuses solely on CSLR tasks, our MixSignGraph can address both CSLR and SLT tasks(including Sign2Text, Sign2Gloss2Text, and GFSLT).
>
> More importantly, in SLT tasks, we find that the current **bottleneck in Gloss-Free SLT** lies in the **continuous feature distribution** of SL videos, which does not align with the **discrete feature distribution** required by translation models, where the inputs are typically discrete tokens (see Figure 4 and Section 3.1). In contrast, gloss-based SLT benefits from gloss annotations with CTC constraints to effectively discretize SL features. Building on this insight, we propose a TCTC pretraining method, which leverages pseudo gloss sequences and CTC to guide the discretization of SL video features.
>
>
>  ## Q2: Insufficient Comparison
>
> Before making a comparison, we sincerely want to state the following important notes.
>
> -  *Contemporaneous comparison* : According to the **NeurIPS 2025 guidelines**[See NeurIPS website CallForPapers], papers published online after March 1st, 2025 are considered "contemporaneous"  i.e., NOT required for comparisons. While (1) Uni-Sign (published at **ICLR 2025, April 24, 2025**),
> (2) C2RL (published on **March 19, 2025**, [See IEEE 10933970]) can be considered contemporaneous.
>
> -  *Possible unfair comparison*:  Uni-Sign utilizes large-scale SL data ( CSL-NEWs and YouTube-ASL) for pretraining, and adopts both video and pose data. In contrast, we do not use any SL-related datasets for pretraining, relying solely on video data. These differences between UniSign and our model may lead to an unfair comparison.
>
>
>
> | DataSet    | ours |ours(Full)  |C2rl| VAP | Uni-Sign*|
> |------------|------|---------|------|----|----------|
> | Phoenix14T |51.14|51.14    |50.96|51.28|-
> | CSL-daily  |49.93| 49.93   |48.21|48.72| 55.30
> | How2Sign   |28.01| 39.46   |27.02|30.27| 49.35
> | OpenASL    |25.71| 42.49   |31.36|42.47| -
>
>
> Nevertheless, we also provide a detailed comparison and show the ROUGE score on the test set of four datasets, as shown in the table.
>
> In the main paper, for Phoenix14T and CSL-daily datasets, we fully trained our MixSignGraph and reported the results. While for the large-scale How2Sign and OpenASL datasets, we used a pretrained I3D for feature extraction (see Section 4: Training Settings) to speed up training. **Importantly, I3D was not trained during the process** and this may lead to lower performances on How2Sign and OpenASL. Now, we further provide the full training results (i.e., ours(Full)) on How2Sign and OpenASL, to show the better performances achieved by our MixSignGraph.
>
> Our model MixSignGraph achieves promising performances across four datasets, especially the model with full training. In regard to the higher performance of Uni-sign, it may be caused by the unfair pretraining on other datasets and the usage of extra pose data, as mentioned before.
>
>
> We will further cite these papers and provide a detailed analysis and discussion in the revised version of the paper.
>
> ## Q3: Semantic Inaccuracy
>
> - The pseudo glosses did introduce some semantic inaccuracies. However, directly generating gloss sequences from text without actual gloss annotations is nearly impossible, and these discrepancies are inevitable. Nevertheless, despite these inaccuracies, the proposed TCTC, with inaccuracies in pseudo glosses, can still effectively discretize video features and significantly improve Gloss-free SLT performance.
>
> - In Table 2 of the main paper (on page 7) (also shown below), we compare SLT performance under three conditions: using ground truth(GT) glosses (w/gloss), using pseudo glosses (TCTC), and not using pseudo glosses (w/o TCTC). The results indicate that, despite these inaccuracies, pretraining with pseudo glosses significantly reduces the performance gap compared to gloss-based SLT, demonstrating the effectiveness of TCTC.
>
> - **We also conducted detailed experiments to analyze how different levels of semantic discrepancies affect SLT performance.** First, we performed an ablation study on the proposed TCTC, including experiments with punctuation removal (TCTC w/o RP) and lemmatization removal (TCTC w/o LE). Second, we simulated different levels of semantic discrepancies by randomly corrupting the GT gloss sequence, i.e., using deletion, addition, replacement, and swapping at varying percentages. As shown in the table below, even under highly semantically inaccurate conditions (i.e., when glosses are highly corrupted up to 80%), the SLT performance still outperforms the baseline (w/o TCTC), demonstrating the effectiveness of TCTC.
>
> |Method|Rouge|BLEU1 |BLEU4|WER|Del/Ins|
> |-|-|-|-|-|-|
> | w/ gloss   | 53.84 |54.90 |28.97| 19.01| 4.9/2.99|
> | w/o TCTC |34.56|35.31 |9.40|-|- |
> | w/ TCTC    |51.14| 50.01| 24.02| 59.55 |35.9/2.7|
> |||||||||
> |TCTC w/o RP|50.01| 49.90| 23.14| 61.55 |35.7/3.1
> |TCTC w/o LE|49.14| 49.62| 22.18| 63.55 |37.1/3.5
> |||||||||
> |10% gloss corrupted| 52.98 |53.87 |28.89| 30.01| 15.9/5.0
> |20% gloss corrupted| 51.79 |52.99 |28.01| 39.91| 16.5/6.1
> |40% gloss corrupted|51.01| 49.92| 25.02| 59.53 |55.9/2.7
> |80% gloss corrupted|40.14| 40.62| 20.18| 87.66 |87.1/3.5
> |100% gloss corrupted|37.63| 39.01| 10.02| 100.00 |99.00/0.3
>
>
> We appreciate your questions and comments. Please let us know for any further questions.

---

> > ### Comment · Area_Chair_5iU3 · 2025-08-05
> >
> > Dear Reviewer MqJV,
> >
> > The authors have responded to the original reviews. Could you read the rebuttal and share your thoughts? Does it address your original concerns? Are there any remaining questions for the authors?
> >
> > Best,
> > AC

---

> > ### Comment · Reviewer_MqJV · 2025-08-05
> >
> > Thank you for the response. The authors have addressed some of my concerns.
> >
> > However, I would like to clarify that the papers I mentioned were released within a reasonable timeframe.
> > C2RL was first posted on arXiv on August 19, 2024, and
> > Uni-Sign was first posted on January 25, 2025.
> >
> > Regarding the comparisons, while ROUGE is reported, BLEU-4 is generally considered a more meaningful metric for translation quality. The current comparisons lack substance—particularly against VAP—and the claim that “our model surpasses the SOTA models” is not well supported.
> >
> > Furthermore, if the main contribution is an improvement on the visual representation side, then isolated sign recognition or continuous sign recognition tasks would be more appropriate targets, where the benefits can be more directly evaluated. If the goal is to demonstrate the effectiveness of TCTC in gloss-free SLT, then fair and rigorous comparisons with relevant SOTA models are necessary.
> >
> > For these reasons, I will update my score accordingly.

---

> ### Author Response · Authors · 2025-08-06
>
> Thanks for taking the time to review our responses.
> ## Q1: Comparisons with ArXiv Preprints
> - Thanks for providing the information about arXiv preprints. While we appreciate the value of arXiv preprints, the arXiv preprints have not yet undergone peer review and may be subject to further revisions. For consistency and fairness, we mainly compare with officially published work  (conference or journal papers). Nevertheless, we are happy to cite and analyze these papers in the final version.
>
> - We further provide the detailed comparisons with VAP, C2RL, and UniSign on Gloss-free SLT (GFSLT) in Table A1-A4. **(1)** VAP adopts a three-stage pretraining with contrastive, language modeling, and CTC losses, and leverages fine-grained pose data for GFSLT.  **(2)** C2RL uses both contrastive and language modeling losses for backbone pretraining.  **(3)** UniSign leverages external SL datasets (CSL-News, YouTube-ASL) for pretraining and adopts both pose and cropped hand regions. In contrast, our method uses only video data without any external SL datasets for pretraining, thus the comparison between UniSign and ours may be potentially unfair. **(4)** We employ a single TCTC loss (CTC-based) for pretraining, and train the model in half-precision due to limited GPU memory (see Sec. 4).
>
> - **1)** UniSign achieves best performance on three datasets, due to additional pretraining and pose data. While for VAP, C2RL, and MixSignGraph, no single model achieves the best performance across all datasets and metrics.  **2)** Comparison with VAP and C2RL: On PHOENIX14T, we achieve lower scores than VAP and slightly lower scores on some metrics than C2RL. On CSL-Daily, we achieve comparable scores with VAP and C2RL. On How2Sign, we outperform both VAP and C2RL. On OpenASL, we achieve lower scores on some metrics than VAP and outperform C2RL.
>
> **Table A1: GFSLT Comparsion on Phoenix14T.**
> |Phoenix14T|R|B1|B2|B3|B4
> |-|-|-|-|-|-|
> |VAP|51.28|53.07|||26.16
> |C2RL|50.96 |52.81|40.20|32.20|26.75
> |UniSign*||||||
> |Our|51.14 |50.01|38.04|29.95|24.02
>
> **Table A2: GFSLT Comparison on CSL-Daily**
> |CSL-Daily|R|B1|B2|B3|B4
> |-|-|-|-|-|-|
> |VAP|48.56| 49.99|||20.85
> |C2RL|48.21 |49.32| 36.28| 27.54| 21.61
> |UniSign*|56.51|55.08 |||26.36
> |Our|49.93| 50.24|36.91| 27.54| 20.78
>
> **Table A3: GFSLT Comparsion on How2Sign**
> |How2Sign| R|B1|B2|B3|B4
> |-|-|-|-|-|-|
>  |VAP|27.77|39.22|||12.87
> |C2RL|27.02|29.07|18.56|12.92|9.37
> |UniSign*|43.22|49.35|||23.14
> |Our(I3D)|28.01 |34.74|20.83 |14.41| 10.41
> |Our(Full)|39.46|39.56|29.34|20.56|14.6
>
> **Table A4: GFSLT Comparison on OpenASL**
> |OpenASL|R|B1|B2|B3|B4
> |-|-|-|-|-|-|
>  |VAP|41.38|45.92|||21.23
> |C2RL|31.36| 31.46|21.85|16.58|13.21
> |UniSign*|43.22|49.35|||23.14
> Our(I3D)|25.7|25.71 |26.65| 16.55 |11.68| 8.69
> Our(Full)|42.49|39.61|29.68|20.39|15.61
>
> ## Q2: Description of SOTA performance
>
> - Thank you for the reminder. In the abstract, we state that “our model surpasses the SOTA models on multiple sign language tasks across several datasets,” which is not intended to claim superiority across all datasets or tasks. Instead,  we mean that our method achieves good results on multiple tasks and datasets, i.e., covering most—but not all—metrics.
>
> - After being kindly requested to compare with recent methods, we did not repeat this statement in the rebuttal. As shown in our updated comparisons, our method achieves comparable performance with these new baselines.
>
> - We fully acknowledge the importance of rigorous description and will revise the description regarding ‘our model surpasses the SOTA models’.
>
> ## Q3: For visual representation contribution, ISR or CSLR may be more appropriate. For TCTC's effectiveness, fair and rigorous comparisons are essential.
>
> - Our contributions lie in both (1) enhancing SL-related visual representations via HSG, and (2) improving gloss-free SLT performance through TCTC. Our visual module can benefit both CSLR and SLT by improving SL-specific features. We evaluate our visual module across CSLR and SLT tasks (Sign2Text, Sign2Gloss2Text, GFSLT) and demonstrate its effectiveness.
>
> - For visual representation, **we did report CSLR results** in Tables 3 and 4, MixSignGraph achieves promising WERs of 25.0, 17.3, and 19.0 on CSL-Daily, PHOENIX14, and PHOENIX14T. We also compare with UniSign in Table A5, our model even outperforms UniSign despite its use of external pretraining data, confirming the strength of our visual module.
>
> - For the effectiveness of TCTC, we compare against baseline methods in Tables 5–8 and with VAP, C2RL, and UniSign in Tables A1–A4 across four GFSLT datasets. In Tables 5-8, our model achieves good performance, surpassing existing methods on most metrics. In Tables A1-A4, our model achieves comparable performance to recent methods, showing the effectiveness of TCTC.
>
> **Table A5: CSLR Comparison on CSL-daily.**
> | CSL-Daily | DEV WER  $\downarrow$| Test WER $\downarrow$
> |-|-|-|
> |UniSign*| 26.7|26.0
> |Our|**25.1**|**25.0**
>
> We hope our response can address your concerns.

---

### Note · Authors · 2025-08-13

Dear ACs and Reviewers,

We sincerely thank you for your time and effort.

After our responses, reviewers **rodL**, **yMHm**, and **zjcq** expressed a **positive attitude** toward this paper, and acknowledged that we have addressed their concerns.

Reviewer **MqJV** raised concerns regarding the comparisons with recent models (including two contemporaneous papers that were previously posted on arXiv but formally published after March 1st, 2025) and the main objective of this paper (i.e., HSG or TCTC). We have carefully addressed the concerns, by providing additional comparisons across multiple metrics, making the performance descriptions more rigorous, and clarifying that our contributions lie both in enhancing SL-related visual representations via HSG and in improving gloss-free SLT performance through TCTC. We have not received any further discussions following our responses and sincerely hope that our responses have resolved Reviewer **MqJV**’s concerns.

Aside from these, the remaining revisions are minor and do not affect the core contributions of this paper. We will incorporate all necessary changes into the camera-ready version and sincerely hope our work will be accepted.

Thank you again for your time and consideration.

Sincerely,

Authors

---

### Decision · Program_Chairs · 2025-09-17

**Decision:**

Accept (poster)

**Comment:**

This paper proposes MixSignGraph, an approach for sign language recognition and translation. The key contribution is a hierarchical sign graph that integrates features at multiple levels of granularity, enabling the model to capture both local and global structures. Initially, the paper received one Accept, one Borderline Accept, and two Borderline Reject ratings. The reviewers acknowledged the presence of several novel components in the architecture that set the work apart from prior approaches. Some reviewers also praised the extensive evaluation, although others felt it was insufficient and suggested expanding the experiments to include additional baselines and metrics. In their rebuttal, the authors provided clarifications and additional results, which slightly shifted the picture to one Reject, one Accept, and two Borderline Accept ratings. While the reviews remain divided, the paper received sufficient support from the reviewers, and the AC concludes that there is no strong basis for rejection. The authors are encouraged to carefully address any remaining reviewer feedback when preparing the final version. In particular, they should expand and refine the quantitative evaluation, improve clarity around the metrics, and incorporate any additional promised results, analyses, or discussions from the rebuttal.